# DA-MERGELORA: HYPERNETWORK-BASED LORA MERGING FOR FEW-SHOT TEST-TIME DOMAIN ADAPTATION

## ABSTRACT

Few-shot Test-Time Domain Adaptation (FSTT-DA) seeks to adapt models to novel domains using only a handful of unlabeled target samples. This setting is more realistic than typical domain adaptation setups, which assume access to target data during source training. However, prior FSTT-DA approaches fail to effectively leverage source domain-specific knowledge, relying on shallow batch normalization updates, prompt-based methods that treat the model as a black box, or ensembling strategies that do not capture cross-domain relationships. To address these limitations, we introduce a new FSTT-DA framework that integrates LoRA fine-tuning with model merging. In our approach, separate LoRA modules are fine-tuned on CLIP's vision encoder for each source domain. Since LoRA modifies only a small fraction of the model's parameters, it retains the base model's generalized knowledge while internally learning domain-specific features. To adapt the learned knowledge to a specific target domain, we propose a hypernetwork trained via meta-learning that generates per-column merging factors to combine LoRA modules. Given a small batch of target images, the hypernetwork produces merging weights that fuse source LoRA modules into a single adapted representation. Our results demonstrate state-of-the-art performance across various domain adaptation datasets. Our source code and trained models will be available upon acceptance of the paper.

## 1 INTRODUCTION

Machine learning models typically assume that training and test data are identically distributed; when the test distribution differs (domain shift), performance degrades (Wang et al., 2022; Zhou et al., 2022). Test-Time Adaptation (TTA) addresses domain shift by adapting a model at inference time, typically operating in either an offline or online mode. In the offline setting, the model is given a large corpus of unlabeled target data and performs multi-step unsupervised adaptation before inference. In the online setting, the model receives a stream of target samples and updates itself incrementally, often on a per-sample or per-mini-batch basis, throughout deployment (Wang et al., 2021). Both modes assume access to many target samples and allow repeated adaptation steps.

In contrast, Few-Shot Test-Time Domain Adaptation (FSTT-DA) assumes that the model receives only a single small batch of unlabeled target images at deployment and must perform a one-shot adaptation step before processing the entire target domain, with no further updates (Wu et al., 2024b). In this setting, the model is trained on labeled data from multiple source domains during development, but the target domain remains completely unseen until test time and is available only as a small unlabeled batch for adaptation. This differs from typical multi-source domain adaptation (Peng et al., 2019), which assumes access to the unlabeled target domain during source training for alignment or distribution matching.

FSTT-DA naturally arises in real applications where data are limited, inference-time overhead must be minimal, or adaptation using target data is restricted by privacy or safety constraints. For example, medical imaging models deployed in new hospitals face domain shifts; however, regulations can prevent iterative or large-scale adaptation using patient data (Bandi et al., 2018). Likewise, robots entering novel environments typically receive only a handful of initial observations before they must

begin downstream tasks. FSTT-DA is therefore a practical yet more challenging setting than typical TTA.

This setting posing two key difficulties: **(CI)** extracting domain-specific cues from diverse source domains without harming cross-domain generalization, and **(CII)** transferring that knowledge to a new target domain under severe data and label scarcity. Despite recent progress, current FSTT-DA methods remain limited. BatchNorm-based adaptation, MABN (Wu et al., 2024b), is efficient but restricted to re-estimating BatchNorm statistics and affine parameters; with small test batches, these estimates become noisy and lead to unstable transfer. Meta-DMoE (Zhong et al., 2022) employs ensemble/distillation strategies to train multiple source-domain experts via a teacher–student pipeline, incurring substantial compute, scaling poorly with the number of sources, and limiting cross-domain sharing. Prompt-based approaches (Chi et al., 2024; 2025) treat the backbone model as a black box and adjust only external prompts, leaving internal representations fixed and restricting knowledge transfer. As a result, existing methods make shallow updates, scale poorly, or restrict knowledge sharing, and ultimately fall short of the two key requirements **(CI–CII)**.

To overcome **(CI)**, we draw inspiration from recent work showing that inserting lightweight low-rank adapters (LoRA) (Hu et al., 2022) in parallel to model weights can effectively capture domain-specific information (Liu et al., 2024). We attach separate sets of LoRAs to a frozen CLIP backbone to model knowledge from multiple source domains. This design provides more expressive adaptation than shallow BatchNorm updates (Wu et al., 2024b) and allows internal knowledge steering, unlike black-box prompting methods such as VDPG (Chi et al., 2024) and L2C (Chi et al., 2025). Moreover, unlike ensemble methods that require a full model per source domain, our approach enables cross-domain knowledge sharing while retaining the generalization of the base model.

To address **(CII)**, we adopt parameter-space model merging to integrate domain-specific LoRA modules into a single model adapted to the target domain. Unlike prompt-based methods (Chi et al., 2024; 2025), merging updates internal weights and aligns both low- and high-level features. Compared to ensembling (Zhong et al., 2022), it avoids running multiple experts and produces one model with fixed memory and a single forward pass. Beyond efficiency, merging promotes direct knowledge sharing across domains and avoids the instability of retraining or distillation in the few-shot setting.

To further motivate our approach, parameter-space model merging has been successfully applied in several other areas. One line of work merges task-specific models into a single multi-task generalist model (Yadav et al., 2023; Jin et al., 2022; Matena & Raffel, 2022; Marczak et al., 2025; Stoica et al., 2024; Gargiulo et al., 2025). Unlike these approaches, which aim to unify different tasks into one model, our goal is to create a specialized domain-specific model tailored to a novel visual domain. Another line of work merges pairs of target LoRAs at test time to produce mixed style-content LoRAs in diffusion models (Shenaj et al., 2024; Shah et al., 2024). In contrast, our setting has no target LoRA available; instead, we must synthesize a domain-specific LoRA by combining source-domain LoRAs guided by only a few unlabeled target samples. Unlike diffusion-based hypernetwork approaches, our hypernetwork accepts two modalities: the LoRA models (as in prior work) and a batch of target images that guide the merge. Model merging has also been explored for NLP task adaptation (Wu et al., 2024a; Huang et al., 2024; Akiba et al., 2025), where each LoRA corresponds to a different supervised task (QA, translation, NLI, etc.). Our setting keeps the task fixed (classification with the same label space) while the input distribution shifts across visual domains. Domain adaptation requires solving the same task under new visual styles and environments, fundamentally differing from previous task-merging scenarios and motivating our domain-driven merging strategy.

To this end, we introduce **DA-MergeLoRA**, an FSTT-DA framework that reformulates adaptation as parameter-space merging. For each source domain, we fine-tune a separate LoRA module on the CLIP vision encoder (Radford et al., 2021), while keeping the base parameters frozen to preserve cross-domain generalization. At test time, given a few unlabeled target images, a hypernetwork predicts merging weights to combine the source-domain LoRAs into a single target LoRA. The hypernetwork is conditioned on both target images and the pool of source LoRA weights, and is trained in a meta-learning fashion to learn effective merging policies. Empirically, our approach achieves state-of-the-art results across multiple FSTT-DA benchmarks, including a **+1.24%** gain in average accuracy on DomainNet, **+2.70%** on Camelyon17, and **+11.40%** in worst-case accuracy on FMoW.

Our work makes the following contributions: (i) Novel FSTT-DA framework: we formulate FSTT-DA as a parameter-space merging problem, yielding a single adapted model. (ii) Hypernetwork architecture: We introduce a meta-learned hypernetwork that generates per-column merge weights to combine multiple source LoRA modules into a specialized, domain-specific target model, conditioned on a small batch of unlabelled target data, enabling effective FSTT-DA. (iii) State-of-the-art results: Our approach achieves SOTA on the DomainNet and WILDS benchmarks, empirically demonstrating that parameter-space merging is an effective approach for FSTT-DA. Together, these contributions move beyond prior FSTT-DA methods (batch normalization, prompt-based, and distillation ensembles) toward modular internal source-domain representations and an adaptive parameter-space mechanism for effective test-time adaptation.

## 2 RELATED WORK

**Test-time domain adaptation.** Distribution shift between training and test data often degrades performance when models are deployed to new domains (Csurka et al., 2017). Domain adaptation algorithms address this by learning features shared between source and target domains, enabling better generalization (Long et al., 2015; Ganin et al., 2016; Bousmalis et al., 2016). Test-time adaptation (TTA) considers the setting where no target-domain data is available during model development (Wang et al., 2021). After deployment to a new domain, the model may adapt in an offline manner, where the model is trained on all unlabelled target data before inference, or in an online manner, where the model is continually adapted to streaming test data (Liang et al., 2025). Common TTA strategies include entropy minimization (Wang et al., 2021; 2024), pseudo-labels (Liang et al., 2020), auxiliary tasks (Sun et al., 2020; Liu et al., 2022), and contrastive learning (Chen et al., 2022).

**Few-shot test-time domain adaptation.** In FSTT-DA, models adapt to unseen domains using only a batch of unlabeled samples at inference (Zhong et al., 2022). Prior works approach this challenge in different ways. Meta-DMoE distills knowledge from a mixture of source-domain experts to a new target model via a meta-learned Transformer aggregator (Zhong et al., 2022). MABN adapts batch-normalization affine parameters using a self-supervised auxiliary branch (Wu et al., 2024b). VDPG generates domain-specific visual prompts from a knowledge bank conditioned on target samples (Chi et al., 2024). L2C enhances VDPG by attaching a parallel branch to CLIP, learning directly from dataset-specific input knowledge and text semantics (Chi et al., 2025).

**Foundation models.** Recent FSTT-DA methods,VDPG (Chi et al., 2024) and L2C (Chi et al., 2025), use CLIP as a frozen backbone. CLIP is a vision–language model whose image and text encoders are trained using contrastive learning so that paired images and captions have high cosine similarity. Classification then selects the prompt "a photo of a <classname>" with the highest similarity (Radford et al., 2021). Vision–language models are appealing for domain adaptation because of their strong out-of-distribution generalization. However, prior research shows that full fine-tuning of CLIP can degrade its generalization capabilities (Wortsman et al., 2022b), motivating parameter-efficient methods such as prompt tuning (Chi et al., 2024) and LoRA (Hu et al., 2022), which keep the backbone frozen while learning domain-specific features. Similar to VDPG and L2C, our work builds on CLIP, but differs by encoding domain knowledge directly in LoRA modules, enabling richer and more flexible adaptation than prompt- or batch normalization-based approaches.

**Model merging.** Model-merging methods aim to combine multiple task-specific models into a single general-purpose model while mitigating task interference. Examples include parameter averaging (Wortsman et al., 2022a; Choshen et al.), Task Arithmetic (Ilharco et al., 2022), Fisher Merging (Matena & Raffel, 2022), RegMean (Jin et al., 2022), and structured merging methods such as TIES (Yadav et al., 2023), which resolves sign conflicts and averages only aligned updates. Several recent advances further improve alignment and reduce interference: KnOTS (Stoica et al., 2024) applies singular value decomposition (SVD) to concatenated LoRA updates before merging task-aligned components; Iso-C / Iso-CTS (Marczak et al., 2025) use SVD and isotropic scaling to identify shared and task-specific directions; and TSV-M (Gargiulo et al., 2025) extracts low-rank task directions via SVD and orthogonalizes them across tasks. These approaches all aim to produce a single multi-task model that preserves performance across the original training tasks. In contrast, our method targets few-shot test-time domain adaptation, where the objective is to generate a domain-specialized model for a previously unseen target domain. Merging in prior work is unconditional (merge rules do not depend on target data) whereas our merging is conditional and meta-learned, driven by a target-domain embedding extracted from a few unlabeled target images.

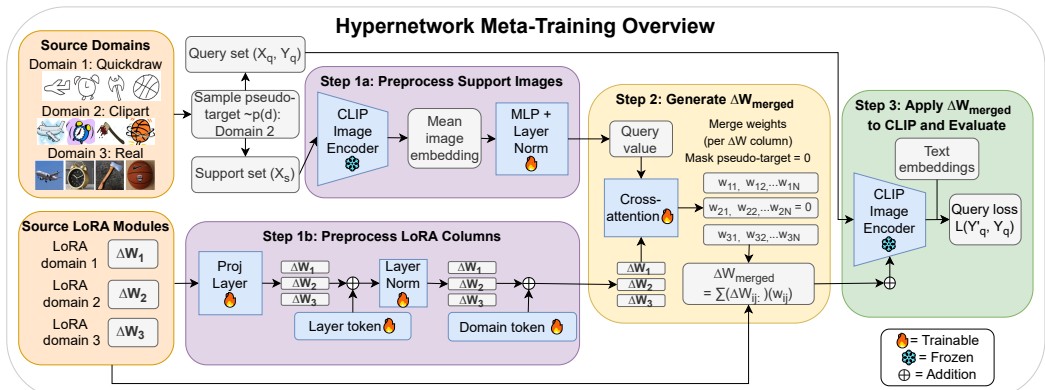

Figure 1: Overview of our hypernetwork training procedure – A LoRA module is trained on each source domain using a frozen CLIP backbone. Afterwards, a meta-trained hypernetwork learns to generate merge weights to combine the columns from each LoRA $\Delta W$, conditioned on pseudo-target support images. The merged LoRA model is applied to the frozen backbone and evaluated with pseudo-target query images and labels. LoRA columns from the pseudo-target domain are masked during training to simulate adaptation to unseen domains.

**LoRA Merging.** LoRA fine-tuning inserts low-rank adapters into frozen foundation models, enabling efficient adaptation with fewer trainable parameters. LoRA merging has recently been applied across multiple domains (Huang et al., 2024; Shenaj et al., 2024; Wu et al., 2025; Shao et al., 2025; Shu et al., 2025; Li et al., 2025; Lee et al., 2024; Charakorn et al., 2025). In diffusion models, LoRA.rar (Shenaj et al., 2024) and ZipLoRA (Shah et al., 2024) merge pairs of content and style LoRAs for personalized image generation, while MoLE (Wu et al., 2024a) trains a gating network to combine multiple content-specific LoRAs into a mixed-content adapter. These approaches all assume that target LoRA modules already exist and focus on fusing them coherently. In contrast, we assume no target LoRA is available; our method must construct a target LoRA by merging source-domain LoRAs, guided only by a few unlabeled target images. Several recent works explore LoRA merging and LoRA generation in NLP and vision. LoRAHub (Huang et al., 2024) learns to merge LoRA modules for NLP task transfer but requires few-shot labeled examples, whereas our adaptation uses only unlabeled target data. SD-LoRA (Wu et al., 2025) incrementally incorporates new LoRA modules for continual learning by decoupling their direction and magnitude. Text-to-LoRA (Charakorn et al., 2025) uses a hypernetwork to generate LoRA weights from natural-language task descriptions. ICM-Fusion (Shao et al., 2025) fuses LoRA modules using a VAE-based latent representation, and RegCL (Shu et al., 2025) merges LoRA adapters for continual domain-specific segmentation. These diverse successes from other research areas further motivate exploring LoRA merging within the FSTT-DA setting.

**Meta-learning.** Meta-learning is a training paradigm that enables a model to quickly adapt to new tasks (Finn et al., 2017). Previous FSTT-DA methods, such as VDPG (Chi et al., 2024), use meta-learning to quickly adapt to novel domains using a prompt generator. Following a similar approach, we use meta-learning to train the hypernetwork so it can rapidly adapt to the target domain.

## 3 METHOD

**Problem setting.** In this work, we follow the same problem setting as previous FSTT-DA methods (Zhong et al., 2022; Chi et al., 2024; 2025). The training set consists of $N$ labeled source domains $\mathcal{D}_s = \{\mathcal{D}_s^n\}_{n=1}^N$, where each source domain $\mathcal{D}_s^n = (x_s, y_s)^n$ contains image–label pairs $(x, y)$. The test set includes $M$ target domains $\mathcal{D}_t = \{\mathcal{D}_t^m\}_{m=1}^M$, where each target domain $\mathcal{D}_t^m = (x_t)^m$ provides only unlabeled images $x_t$. We assume a distribution shift exists between any source and target domain, while all domains share the same label space. FSTT-DA trains exclusively on the source domains, without access to target data. At test time, when an unseen target domain $\mathcal{D}_t^m$ is encountered (deployed in a new environment), the model must adapt to this domain using only a few-shot set of unlabeled samples. The adapted model is then evaluated on the entire target domain.

**Overview.** In this work, we develop a hypernetwork that generates merging factors to combine LoRA modules, conditioned on a batch of target images. Sec. 3.1 outlines the training of individual LoRA modules. Sec. 3.2 presents the architecture of the LoRA hypernetwork, which produces per-column merging weights. Sec. 3.3 details the meta-learning setup used to train the hypernetwork and the inference procedure.

## 3.1 LoRA Module Training

To capture domain-specific knowledge, a LoRA model is trained on each source domain using CLIP (Radford et al., 2021). LoRA adapts the vision encoder by adding trainable low-rank matrices to its attention and projection weight matrices (Hu et al., 2022). Instead of fine-tuning the weight matrix $W \in \mathbb{R}^{d \times k}$, LoRA replaces the weight update with a low-rank approximation, learned through two smaller matrices $A$ and $B$:

$$W' = W + \Delta W, \quad \Delta W = BA \tag{1}$$

where $A \in \mathbb{R}^{r \times k}$ and $B \in \mathbb{R}^{d \times r}$ with $r \ll \min(d, k)$. The rank $r$ controls the trade-off between capacity and efficiency. During training, only $A$ and $B$ are updated, while $W$ remains frozen. For an input $h$, the forward pass becomes:

$$h' = Wh + BAh = Wh + \Delta Wh. \tag{2}$$

The backbone CLIP model is frozen throughout training, and only the LoRA parameters are updated. Before training, class-specific text prompt embeddings (e.g., "a photo of a <classname>") are precomputed using CLIP's text encoder. During training, the model uses the cross-entropy loss to learn to select the prompt embedding with the highest cosine similarity to the LoRA-modified image embedding.

## 3.2 LoRA Merging Hypernetwork

A hypernetwork is a neural network that generates the weights or parameters of another network (Shenaj et al., 2024). Here, we use a hypernetwork that learns to generate merging weights to combine the columns of LoRA matrices $\Delta W$ trained on different source domains. It accepts as input the batch of target images and the LoRA columns from the different source domains. The hypernetwork uses cross-attention to find the LoRA columns from the domains which correlate best with the target domain images.

**Target image preprocessing:** Before being used for cross-attention, the batch of target domain images is preprocessed. The images, $X_d$, are passed through the frozen CLIP backbone to generate embeddings, and the mean embedding, $e_d$, is found:

$$e_d = \text{mean}(\text{EncodeImage}(X_d)) \tag{3}$$

The mean embedding is then passed through a small multi-layer perceptron (MLP) network to reduce its dimension to $\mathbb{R}^{128}$ and extract domain-specific information. The MLP consists of two linear layers with a GELU activation and hidden dimension size of $\mathbb{R}^{256}$. The output from the MLP is then passed through a LayerNorm (LN) to normalize the output:

$$e'_d = \text{LN}(\text{MLP}(e_d)) \tag{4}$$

**LoRA column preprocessing:** Before being used for cross-attention, the LoRA columns are also preprocessed. Firstly, they are passed through a projection layer, which maps their input dimension to a uniform dimension of $\mathbb{R}^{128}$. Additionally, we add three content tokens to the columns to help the hypernetwork learn more fine-grained and context-aware merging weights: (i) Layer index token: a learnable embedding $u_\ell \in \mathbb{R}^{128}$ representing the transformer layer $\ell$ the column came from; (ii) Sub-layer type token: a learnable embedding $v_t \in \mathbb{R}^{128}$ indicating whether the column comes from an attention (`qkv`) or projection (`proj`) site of the transformer layer. These two tokens act as positional embeddings. (iii) Domain token: for each source domain $d$, a learnable embedding $q_d$ is projected to $\mathbb{R}^{128}$ and added after normalization, providing a lightweight bias indicating which domain the column came from:

$$c^\star = \text{LN}(\tilde{c} + u_\ell + v_t) + \alpha P_{\text{dom}} q_d, \tag{5}$$

Here, $\tilde{c} \in \mathbb{R}^{128}$ is the projected LoRA column, $P_{\text{dom}} \in \mathbb{R}^{128 \times E'}$ projects the domain embedding into the 128-dimensional space, and $\alpha$ is a small scaling factor that controls the influence of the domain token. The resulting $c^\star \in \mathbb{R}^{128}$ is the final context-enriched column representation used in cross-attention.

**Cross-attention:** Cross-attention is then used to produce the merge weights, where the query is the embedded target domain representation and the keys are the processed LoRA columns:

$$w_{\ell,t,c}^{(k)} = \text{softmax}_k \left( \frac{Q \, K_{\ell,t,k,c}^\top}{\tau} \right), \tag{6}$$

Here, $Q$ is the projected target-domain embedding, $K_{\ell,t,k,c}$ is the key vector for the LoRA column at transformer layer $\ell$, site $t$, domain $k$, and column index $c$, and $\tau$ is a temperature scaling factor. The softmax is taken along the domain axis $k$, producing normalized per-domain weights for each column.

After the merge weights are generated using the hypernet, the LoRA matrices $\Delta W$ from the different source domains are combined together:

$$\left( \Delta W_{\ell,t}^{\text{merge}} \right)_{:,c} = \sum_{k=1}^{K} w_{\ell,t,c}^{(k)} \left( \Delta W_{\ell,t}^{(k)} \right)_{:,c}, \qquad c = 1, \ldots, C_{\ell,t}.$$

Here, $\Delta W_{\ell,t}^{(k)}$ denotes the LoRA update at transformer layer $\ell$ and site $t \in \{\text{attn}, \text{proj}\}$ for source domain $k$. $w_{\ell,t,c}^{(k)}$ is the merging weight for column $c$. $C_{\ell,t}$ is the number of columns at that $(\ell, t)$ site. $K$ is the number of source domains. We follow LoRA.rar (Shenaj et al., 2024) and use per-column merging, rather than per-model or per-layer merging, to allow for fine-grained feature-level merging. After each merged $\Delta W$ is computed, they are applied to the frozen CLIP model to generate the new merged LoRA model.

### 3.3 HYPERNETWORK META-TRAINING AND INFERENCE

**Training:** The hypernetwork is trained using a meta-learning setup, similar to the approach proposed in (Chi et al., 2024). At each training iteration, one source domain is randomly selected as the pseudo-target domain. The LoRA columns of this pseudo-target model are masked with zeros before being input to the hypernetwork, while all other source LoRA models are provided as inputs. From the pseudo-target domain, one batch of images is randomly selected as the support set and another as the query set. The support set conditions the hypernetwork: its average image embedding is passed to the hypernetwork along with the LoRA columns from the other source domains. The hypernetwork then outputs weights for merging the LoRA columns from different source domains. The merge weights of the pseudo-target domain are set to zero. This simulates the process of encountering novel domains at test time. The merged LoRA matrices are applied to the base CLIP model, and the new LoRA model is evaluated using the images and labels from the query set. Cross-entropy is used as the loss; the merged LoRA model selects the precomputed text prompt with the highest similarity to the adapted target image embedding as the predicted label. Gradients flow end-to-end from the predicted outputs of the merged LoRA model back to the hypernetwork. All parameters are frozen except those of the hypernetwork. The training algorithm is summarized in Algorithm 1. Fig. 1 shows the overall training process and hypernetwork architecture.

**Inference:** At test time, a batch of images from the target domain and all source LoRA modules are passed to the hypernetwork. The target domain is novel and completely excluded from the hypernetwork training process. The hypernetwork outputs the merging weights, which are used to combine the source LoRA modules into a new model optimized for the target domain. The adapted model is then evaluated on the entire target domain.

## 4 EXPERIMENTS

**Datasets.** We evaluate on three WILDS (Koh et al., 2021) benchmark classification datasets: iWild-Cam (Beery et al., 2021), FMoW (Christie et al., 2018), and Camelyon17 (Bandi et al., 2018).

---

**Algorithm 1:** Overview of our hypernetwork meta-training algorithm.

---

**Input:** $\mathcal{D}_s$: source domains; $\{\Delta W^{(k)}\}_{k=1}^K$: frozen LoRA weight matrices from the K source domains; frozen base weights $W$ (CLIP); trainable hypernetwork $\mathcal{H}$; step size $\alpha$

**Output:** Trained hypernetwork $\mathcal{H}$

Randomly initialize $\theta$ (params of $\mathcal{H}$)

**while** *not converged* **do**

    Sample pseudo-target domain index $d \sim \{1, \ldots, K\}$

    Sample support/query from domain $d$: $(X_{S_d}, Y_{S_d}), (X_{Q_d}, Y_{Q_d})$

    Compute support embedding with frozen CLIP: $e_{S_d} \leftarrow \mathrm{mean}(\mathrm{EncodeImage}(X_{S_d}))$

    Set pseudo-target LoRA columns to zero: $\Delta W^{(d)} \leftarrow 0$

    Generate per-column merge weights using the hypernet: $w \leftarrow \mathcal{H}\left(e_{S_d}, \{\Delta W^{(k)}\}_{k=1}^K\right)$

    **foreach** *layer, $\ell$, layer type t, and column c* **do**

        $\left(\Delta W_{\ell,t}^{\mathrm{merge}}\right)_{:,c} = \sum_{k=1}^K w_{\ell,t,c}^{(k)} \left(\Delta W_{\ell,t}^{(k)}\right)_{:,c}$

    **end**

    Apply merged LoRA matrices to base CLIP model: $W' \leftarrow W + \Delta W^{(\mathrm{merge})}$

    Evaluate on query set with merged model: $\hat{Y}_{Q_d} \leftarrow f_{W'}(X_{Q_d})$

    Calculate loss: $\mathcal{L} \leftarrow \mathrm{CE}(\hat{Y}_{Q_d}, Y_{Q_d})$

    $\theta \leftarrow \theta - \alpha \nabla_\theta \mathcal{L}$

**end**

---

These datasets consist of real-world data with distribution shift. We use the official WILDS evaluation metrics for each dataset (average accuracy, F1-score, and worst-case (WC) accuracy) and the official train/test-OOD splits. We also evaluate on DomainNet (Peng et al., 2019), which contains roughly 600k images from 345 classes across six domains. We use the standard leave-one-out protocol for DomainNet, selecting one domain for testing and the remainder as source domains. We follow prior FSTT-DA work (Chi et al., 2024; 2025) and evaluate on the WILDS and DomainNet benchmarks. As our method is designed for classification, we exclude the PovertyMap regression benchmark and leave extending to regression tasks for future work. For all datasets, images are preprocessed using CLIP's standard train and test transforms.

The iWildCam dataset contains 243 source domains, some with very few images. Following Meta-DMoE (Zhong et al., 2022), we reduce them to 10 training domains by clustering. Image embeddings are extracted using the frozen CLIP ViT-B/16 model, averaged per domain, and clustered with K-means. IWildCam also exhibits class imbalance, with many domains containing classes exclusive to that domain. This makes the task more challenging, requiring both domain and class merging. To address this, we do not mask the pseudo-target domain during meta-training, providing a stronger learning signal and ensuring all classes are represented.

**Baselines:** We compare our results against common CNN-based (Sun & Saenko, 2016; Arjovsky et al., 2019; Zhang et al., 2021; Zhong et al., 2022; Wu et al., 2024b; Xu et al., 2020; Blanchard et al., 2011; Nam et al., 2021) and CLIP-based (Chi et al., 2024; 2025; Zheng et al., 2022; Cha et al., 2022) domain adaptation methods, which include recent FSTT-DA algorithms (MABN, Meta-DMOE, VDPG, L2C). As well, we compare against zero-shot CLIP (no fine-tuning).

**Training details.** We use OpenAI's pretrained CLIP ViT-B/16 backbone (Radford et al., 2021). All LoRA models use rank 16, $\alpha = 32$, and dropout $= 0$. Learning rates are $5 \times 10^{-4}$ for LoRA fine-tuning, and either $1 \times 10^{-5}$ or $5 \times 10^{-5}$ for hypernetwork training, with a batch size of 16 for both query and support sets. Models are trained for between 1 and 5 epochs. The text prompts used for training are dataset-specific, following L2C's templates (Chi et al., 2025); the mean embedding from the text prompts is used as the final text embedding. Table 3 in Appendix A.2 summarizes all hyperparameters per dataset and training details.

## 4.1 MAIN RESULTS

Table 1b and Table 1a present the performance of our hypernetwork on DomainNet and WILDs, respectively.

**DomainNet.** Compared to the state-of-the-art (SOTA) ViT-B/16 FSTT-DA method L2C (Chi et al., 2025), our approach improves average accuracy on DomainNet by $1.24\%$. Our model outperforms all methods across all domains except "quickdraw," where it performs slightly lower $(-0.83\%)$ than

MIRO (Cha et al., 2022). Relative to the SOTA CNN-based architecture MABN (Wu et al., 2024b), our method improves average accuracy on DomainNet by 16.94%.

**WILDS.** Compared to the SOTA ViT-B/16 FSTT-DA method L2C (Chi et al., 2025), our approach improves average accuracy by 2.80% on FMoW, 2.70% on Camelyon17, and 0.24% on iWildCam, along with a 11.36% gain in worst-case accuracy on FMoW. Our model outperforms L2C on all metrics except the iWildCam F1-score, where it performs slightly worse ($-1.58\%$). Relative to the SOTA CNN-based architecture MABN (Wu et al., 2024b), our method improves average accuracy by 4.40% on FMoW, 4.50% on Camelyon17, and boosts worst-case accuracy on FMoW by 15.66%, while underperforming only on iWildCam. One contributing factor to the slightly weaker performance on iWildCam may be the disjoint label sets across its source domains. Our meta-training method assumes a shared label space, but many iWildCam source domains contain classes that appear in only one location. Prior work shows that domain adaptation performance degrades when label distributions diverge or when classes are domain-exclusive (Chidlovskii, 2019; Yan et al., 2017; Tan et al., 2020). Since our focus is domain adaptation, extending the framework to explicitly handle class-parameter merging is an important direction for future work.

### 4.2 ABLATION AND ANALYSES

We conduct ablation studies on the WILDS datasets, evaluating LoRA training techniques, merging strategies (per-model, per-layer, per-column), and the impact of different architectural components. We also present the heatmaps of merge weights before and after hypernetwork training. Additional ablations and analyses, including architecture comparisons, t-SNE visualizations, batch-size studies, qualitative examples, merge-weight vs. domain similarity correlation analysis, and computational analysis are provided in Appendix A.3.

**Merging weight heatmaps.** To evaluate whether the hypernetwork learns meaningful merging weights, we plot heatmaps of average per-domain merging weights before and after training. Appendix A.3 Fig. 2 shows results for FMoW (2016 test year) and DomainNet ("sketch" target domain). The x-axis denotes source domains (FMoW: years; DomainNet: image styles), and the y-axis denotes transformer layers. Before training, weights are nearly uniform ($\sim 0.09$ for FMoW, $\sim 0.20$ for DomainNet). After training, the hypernetwork shifts weights towards domains visually similar to the target domain (later years for FMoW, "clipart" for sketch), and down-weights dissimilar and difficult domains (earlier years for FMoW, "quickdraw" for sketch), indicating it learns to emphasize relevant domains.

**LoRA training techniques.** We compare our method against four additional techniques for training LoRA on the source data: (i) LoRA-universal – Train a single LoRA model on all source domains combined into one dataset. This yields a domain-invariant LoRA module, rather than domain-specific modules; (ii) LoRA-average – Combine domain-specific LoRA modules by simple parameter-wise averaging; (iii) LoRA-entropy-average – A simple few-shot method where we use a batch of target images to compute the entropy using each source LoRA model. The inverse entropy (normalized) is used as the merging weight, where lower entropy implies higher weight; (iv) LoRA-TIES and LoRA-TIES-per-column – We implement the TIES (Yadav et al., 2023) merging procedure to merge LoRA modules. We evaluate two variants: one that merges all LoRA delta matrices jointly, and one that merges individual delta columns separately (similar to our approach). We use the same hyperparameter settings recommended in the original paper: trim step K=20%, sign election using resolve = mass, and disjoint merging using merge = dis-mean. Results are shown in Table 2a. Our method outperforms the other LoRA techniques. On harder datasets (iWildCam, FMoW), simple averaging, entropy averaging, and TIES perform markedly worse than the hypernetwork (approximately $\sim 20\%$ decrease on IWildCam F1-score, and between 10% to 20% decrease on FMoW worst-case accuracy). On datasets with fewer source domains and more balanced per-domain classes (Camelyon17), simpler merging methods degrade less ($\sim 2\%$ decrease), suggesting the hypernetwork excels in more complex merging situations. Our model also outperforms the universal LoRA in all cases, highlighting the benefit of domain-specific knowledge.

**Per-model vs. per-layer vs. per-column.** Beyond merging LoRA columns, we compare two additional strategies: (i) Per-model – one merge weight per source model; (ii) Per-layer – one merge weight per layer of each source model. In the hypernetwork architecture, per-model merging averages all LoRA columns before cross-attention, while per-layer merging averages columns within each layer. All source models share the same architecture and training hyperparameters to ensure

Table 1: Evaluation of our method (DA-MergeLoRA) on the (a) WILDS datasets and (b) DomainNet dataset. The best CNN and best ViT methods are bolded separately. Results are reported as mean (standard deviation) over three trials with different random seeds. The $\dagger$ symbol denotes the official dataset metric. Our method achieves SOTA results on two of the three official WILDS metrics and on five of the six DomainNet domains.

| Method | Backbone | Clip | Info | Paint | Quick | Real | Sketch | Avg. |
|---|---|---|---|---|---|---|---|---|
| ERM | | 58.1 (0.3) | 18.8 (0.3) | 46.7 (0.3) | 12.2 (0.4) | 59.6 (0.1) | 49.8 (0.4) | 40.9 |
| Mixup (Xu et al., 2020) | | 55.7 (0.3) | 18.5 (0.5) | 44.3 (0.5) | 12.5 (0.4) | 55.8 (0.3) | 48.2 (0.5) | 39.2 |
| CORAL (Sun & Saenko, 2016) | | 59.2 (0.1) | 19.7 (0.2) | 46.6 (0.3) | 13.4 (0.4) | 59.8 (0.2) | 50.1 (0.6) | 41.5 |
| MTL (Blanchard et al., 2011) | CNNs | 57.9 (0.5) | 18.5 (0.4) | 46.0 (0.1) | 12.5 (0.1) | 59.5 (0.3) | 49.2 (0.1) | 40.6 |
| SegNet (Nam et al., 2021) | | 57.7 (0.3) | 19.0 (0.2) | 45.3 (0.3) | 12.7 (0.5) | 58.1 (0.5) | 48.8 (0.2) | 40.3 |
| ARM (Zhang et al., 2021) | | 49.7 (0.3) | 16.3 (0.5) | 40.9 (1.1) | 9.4 (0.1) | 53.4 (0.4) | 43.5 (0.4) | 35.5 |
| Meta-DMoE (Zhong et al., 2022) | | 63.5 (0.2) | 21.4 (0.3) | 51.3 (0.4) | 14.3 (0.3) | 62.3 (1.0) | 52.4 (0.2) | 44.2 |
| MABN (Wu et al., 2024b) | | **64.2** | **23.6** | **51.5** | **15.2** | **64.6** | **54.1** | **45.5** |
| DoPrompt (Zheng et al., 2022) | ViT-B/16 IMN | 67.6 (0.2) | 24.6 (0.1) | 54.9 (0.1) | 17.5 (0.2) | 69.6 (0.3) | 55.2 (0.5) | 48.3 |
| Zero-shot (Radford et al., 2021) | | 69.9 | 48.2 | 65.4 | 14.5 | 82.3 | 62.5 | 57.1 |
| ERM | | 68.0 (0.1) | 22.5 (0.6) | 46.5 (4.2) | 18.5 (0.9) | 58.7 (2.7) | 52.5 (1.2) | 44.4 |
| MIRO (Cha et al., 2022) | ViT-B/16 | 74.9 (0.2) | 37.1 (0.4) | 59.8 (0.6) | **18.7 (1.2)** | 72.2 (0.2) | 61.2 (0.9) | 54.0 |
| VDPG (Chi et al., 2024) | CLIP | 76.3 (0.2) | 49.3 (0.1) | 67.8 (0.1) | 17.4 (0.2) | 81.5 (0.3) | 66.6 (0.2) | 59.8 |
| L2C (Chi et al., 2025) | | 75.6 (0.1) | 52.1 (0.1) | 69.4 (0.1) | 17.3 (0.2) | 85.5 (0.1) | 67.1 (0.2) | 61.2 |
| **DA-MergeLoRA (ours)** | | **76.66 (0.08)** | **54.49 (0.21)** | **70.91 (0.05)** | 17.87 (0.72) | **85.52 (0.12)** | **69.17 (0.08)** | **62.44 (0.16)** |

(a) Evaluation of our method (DA-MergeLoRA) on the DomainNet dataset using the leave-one-out domain testing protocol.

| Method | Backbone | iWildCam | | Camelyon17 | FMoW | |
|---|---|---|---|---|---|---|
| | | Acc | Macro F1$^\dagger$ | Acc$^\dagger$ | WC Acc$^\dagger$ | Acc |
| ERM | | 71.6 (2.5) | 31.0 (1.3) | 70.3 (6.4) | 32.3 (1.25) | 53.0 (0.55) |
| CORAL (Sun & Saenko, 2016) | | 73.3 (4.3) | 32.8 (0.1) | 59.5 (7.7) | 31.7 (1.24) | 50.5 (0.36) |
| IRM (Arjovsky et al., 2019) | | 59.8 (3.7) | 15.1 (4.9) | 64.2 (8.1) | 30.0 (1.37) | 50.8 (0.13) |
| ARM-CML (Zhang et al., 2021) | CNNs | 70.5 (0.6) | 28.6 (0.1) | 84.2 (1.4) | 27.2 (0.38) | 45.7 (0.28) |
| ARM-BN (Zhang et al., 2021) | | 70.3 (2.4) | 23.7 (2.7) | 87.2 (0.9) | 24.6 (0.04) | 42.0 (0.21) |
| Meta-DMoE (Zhong et al., 2022) | | 77.2 (0.3) | 34.0 (0.6) | 91.4 (1.5) | 35.4 (0.58) | 52.5 (0.18) |
| MABN (Wu et al., 2024b) | | **78.4 (0.6)** | **38.3 (1.2)** | **92.4 (1.9)** | **36.6 (0.41)** | **53.2 (0.52)** |
| Zero-shot (Radford et al., 2021) | | 14.9 | 9.7 | 50.1 | 14.5 | 16.3 |
| VDPG (Chi et al., 2024) | ViT-B/16 | 71.4 (0.2) | 30.1 (0.3) | 93.2 (0.3) | 37.8 (0.5) | 52.7 (0.3) |
| L2C (Chi et al., 2025) | CLIP | 73.4 (0.4) | **35.2 (0.3)** | 94.2 (0.2) | 40.9 (0.4) | 54.8 (0.1) |
| **DA-MergeLoRA (ours)** | | 73.70 (1.49) | 33.62 (1.63) | **96.90 (0.14)** | **52.26 (0.16)** | **57.60 (0.14)** |

(b) Evaluation of our method (DA-MergeLoRA) on the WILDS datasets (IWildCam, Camelyon17, and FMoW) under out-of-distribution (OOD) test conditions.

fair comparison. In theory, per-column merging should outperform others, because its finer granularity allows the hypernetwork to generate more flexible weights. However, empirical results (Table 2b) show that all three strategies achieve similar performance across most datasets ($\sim 1\%$ difference across metrics), except that per-model merging underperforms on IWildCam. The weaker performance of per-column merging likely stems from two factors: (i) Source LoRA modules are trained independently on different domains, therefore columns may not be perfectly aligned. As a result, column-wise merging may combine misaligned information and thus degrade performance (Shah et al., 2024); and (ii) the single-head cross-attention hypernetwork has limited capacity to model fine-grained per-column weights. Future work could address these issues by explicitly aligning columns across source domains and employing more expressive hypernetwork architectures.

**Component analysis.** We remove specific components of the hypernetwork to evaluate their impact on performance. The components examined are: (i) Layer and layer-type tokens – we remove the learnable transformer layer ID token embeddings and layer-type (attention or projection) token embeddings appended to the LoRA columns; (ii) Domain tokens – we remove the learnable domain-specific token embeddings appended to the LoRA columns; (iii) Conditional target images (replaced with noise) – we remove the pseudo-target and target images as inputs to the hypernetwork during training and inference, replacing them with random noise; and (iv) Conditional target images (replaced with a trainable vector) – We remove the target-image embedding branch and replace it with a trainable vector. Instead of using pseudo-target images as the support set during training, we train a learnable query vector, which represents learning a general merging policy. Table 2c reports the results on WILDS. For IWildCam, removing the conditional target images sharply reduces perfor-

Table 2: Ablation experiments on the WILDS datasets. Results are reported as mean (standard deviation) over three trials with different random seeds. The $^\dagger$ symbol denotes the official dataset metric.

| Method | IWildCam Acc | IWildCam Macro F1$^\dagger$ | Camelyon17 Acc$^\dagger$ | FMoW WC Acc$^\dagger$ | FMoW Acc |
|---|---|---|---|---|---|
| LoRA-universal | 72.77 (3.13) | 28.88 (1.12) | 89.81 (6.89) | 51.02 (1.28) | 55.69 (0.24) |
| LoRA-average | 14.98 (0.19) | 10.21 (0.05) | 95.26 (0.12) | 20.63 (0.08) | 21.95 (0.11) |
| LoRA-entropy-average | 41.89 (0.81) | 12.24 (0.22) | 95.28 (0.09) | 21.71 (0.50) | 23.15 (0.25) |
| LoRA-TIES | 49.85 (3.25) | 12.94 (0.84) | 95.95 (0.18) | 38.54 (1.27) | 43.98 (1.06) |
| LoRA-TIES-per-column | 51.25 (3.09) | 13.44 (0.20) | 95.35 (0.45) | 40.34 (1.12) | 45.35 (0.89) |
| **DA-MergeLoRA (ours)** | **73.70 (1.49)** | **33.62 (1.63)** | **96.90 (0.14)** | **52.26 (0.16)** | **57.60 (0.14)** |

(a) Comparison of LoRA training strategies: LoRA-universal (a single LoRA model trained on all source data combined), LoRA-average (simple parameter averaging), LoRA-entropy-average (entropy-weighted averaging), LoRA-TIES (TIES merging applied to all LoRA parameters at once), LoRA-TIES-per-column (TIES merging applied to each LoRA delta column separately), and ours (hypernetwork-based merging).

| Method | IWildCam Acc | IWildCam Macro F1$^\dagger$ | Camelyon17 Acc$^\dagger$ | FMoW WC Acc$^\dagger$ | FMoW Acc |
|---|---|---|---|---|---|
| Per-model | 56.81 (1.44) | 18.39 (0.28) | 96.07 (0.89) | **53.22 (0.10)** | **58.01 (0.32)** |
| Per-layer | 73.64 (1.40) | 33.53 (1.35) | 96.07 (0.72) | 53.03 (0.46) | 57.94 (0.37) |
| **Per-column (ours)** | **73.70 (1.49)** | **33.62 (1.63)** | **96.90 (0.14)** | 52.26 (0.16) | 57.60 (0.14) |

(b) Comparison of per-model merging (one merge weight per source model), per-layer merging (one merge weight per layer for each source model), and per-column merging (one merge weight per column within each layer).

| Method | IWildCam Acc | IWildCam Macro F1$^\dagger$ | Camelyon17 Acc$^\dagger$ | FMoW WC Acc$^\dagger$ | FMoW Acc |
|---|---|---|---|---|---|
| No layer and layer type token | 72.31 (0.43) | 28.95 (0.78) | 96.86 (0.34) | 49.16 (0.34) | 54.79 (0.05) |
| No domain token | 67.21 (1.50) | 25.72 (1.81) | **96.92 (0.12)** | 48.80 (0.38) | 54.16 (0.02) |
| No target images (noise) | 55.86 (1.00) | 17.39 (0.66) | 95.86 (0.48) | 43.04 (0.28) | 48.43 (0.18) |
| No target images (trainable vector) | 63.47 (1.30) | 19.92 (0.65) | **96.92 (0.14)** | 51.98 (0.18) | 57.36 (0.13) |
| **DA-MergeLoRA (ours)** | **73.70 (1.49)** | **33.62 (1.63)** | 96.90 (0.14) | **52.26 (0.16)** | **57.60 (0.14)** |

(c) Ablation studying the effect of: (i) removing the layer and layer-type tokens, which indicate the transformer layer from which each LoRA column originates; (ii) removing the domain tokens, which identify the source domain of each column; (iii) providing random noise as input to the hypernetwork instead of target images for the support set; and (iv) replacing the target-embedding branch with a trainable query vector that learns a general merging policy.

mance (a decrease of 13.7%–16.2% in F1-score). This is expected, as IWildCam is the most difficult dataset, with 48 target domains exhibiting significant domain shifts. For FMoW, removing conditional images also decreases performance, though not as severely. Results show that the learnable query vector reduces performance less than using noise as input, indicating that a general merging policy can still learn useful priors from the source domains. However, as seen in IWildCam, for complex domain shifts, a domain-specific model performs better. Removing the layer tokens and domain tokens also lowers performance for both IWildCam and FMoW, showing that appending layer- and domain-specific information enables the hypernetwork to produce more meaningful merge weights. For Camelyon17, removing any component has little effect, as it is a simpler dataset with only three source domains, where the optimal merge weights are already close to uniform, as shown by the simple parameter-wise averaging results in Table 2a.

# 5 CONCLUSION

In this work, we introduce a model-merging framework for few-shot test-time domain adaptation. We capture domain-specific information by training a LoRA module for each source domain. A hypernetwork, trained via meta-learning, then generates merging weights to optimally combine these modules for adaptation to a target domain, given a batch of unlabeled target images. Our method effectively adapts to novel domains and achieves state-of-the-art performance across diverse benchmarks. This work also opens several avenues for future research, including: (i) exploring merging strategies beyond meta-learning (e.g., reinforcement learning); (ii) developing mechanisms to handle class imbalance and novel classes; (iii) leveraging CLIP's text encoder for stronger vision–language alignment; (iv) increasing the diversity of pseudo-target domains during meta-training through data augmentation, style mixing, or combining images from multiple source domains; (v) designing richer hypernetwork architectures such as bidirectional cross-attention; and (vi) employing LoRA column-alignment strategies to improve merging.

REPRODUCIBILITY STATEMENT

To ensure reproducibility, we follow standard best practices in machine learning. All code will be made publicly available on GitHub upon acceptance of this paper, including configuration files and scripts to reproduce experiments. All hyperparameters and training details are reported in the paper. We also provide hardware and software configurations (GPU type, CUDA/PyTorch version) to facilitate replication. Experiments were run over multiple trials with fixed random seeds. All datasets are open-source and publicly available, and we use the official WILDS library for data loading and evaluation.

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

# A  APPENDIX

## A.1  LARGE LANGUAGE MODEL USAGE DECLARATION

LLMs were used in the development of this paper for the following purposes:

1. To aid/polish writing: ChatGPT was used for proofreading. Example prompt: "Proofread this paragraph for spelling, grammar, and conciseness without changing the meaning."

2. For retrieval and discovery: ChatGPT was used to help identify related work. Example prompt: "What are commonly used model merging techniques?"

3. For research ideation: ChatGPT was used to brainstorm ideas, including clarifying and understanding existing approaches. Example prompt: "Can you explain how this paper's algorithm works?"

4. For code development and debugging: All code was written in Cursor (an IDE based on Visual Studio Code with a built-in LLM) to assist with coding and debugging. Example prompt: "What is causing this bug and how can I fix it?"

All LLM outputs were reviewed by the authors for accuracy. All novel research ideas, scientific contributions, analyses, and results were developed independently by the authors without the use of LLMs.

## A.2  HYPERPARAMETER SELECTION

Appendix Table 3 summarizes the hyperparameters used for each experiment and dataset. For hypernetwork meta-training, one epoch is defined as 1000 episodes, with each episode consisting of a single update using the support and query sets. For the hypernetwork's domain-token scaling factor ($\alpha$), we use 0.1 for all datasets, except for IWildCam, where we use 0.3. All experiments are trained with 6 workers, pinned memory, and the AdamW optimizer (Loshchilov & Hutter, 2017). For all source-LoRA experiments, we use AdamW hyperparameters of betas = [0.9, 0.999], eps = 1e-8, and weight decay = 0.001. For all hypernetwork experiments, we use AdamW hyperparameters of betas = [0.9, 0.999], eps = 1e-8, and weight decay = 0.01. Experiments are implemented in PyTorch 2.8.0 (CUDA 12.8) (Paszke et al., 2019) and run on NVIDIA A100 GPUs with 20GB memory.

Table 3: Hyperparameters used for different experiments across datasets.

| Experiment | Dataset | Learning Rate | # Epochs | Batch Size |
|---|---|---|---|---|
| LoRA (All Sources) | DomainNet | 5e-4 | 1 | 128 |
| | iWildCam | 5e-4 | 1 | 128 |
| | FMoW | 5e-4 | 1 | 128 |
| | Camelyon17 | 5e-4 | 1 | 128 |
| LoRA (Per Domain) | DomainNet | 5e-4 | 1 | 128 |
| | iWildCam | 5e-4 | 5 | 128 |
| | FMoW | 5e-4 | 5 | 128 |
| | Camelyon17 | 5e-4 | 1 | 128 |
| Hypernet (Cross-Attn) | DomainNet | 1e-5 | 1 | 16 |
| | iWildCam | 5e-5 | 2 | 16 |
| | FMoW | 1e-5 | 5 | 16 |
| | Camelyon17 | 1e-5 | 3 | 16 |
| Hypernet (MLP) | iWildCam | 5e-4 | 4 | 16 |
| | FMoW | 1e-5 | 5 | 16 |
| | Camelyon17 | 5e-5 | 1 | 16 |

### A.3 Additional Ablation and Analyses

### A.3.1 Merging Weight Heatmaps

Appendix Fig. 2 provides the full set of merging-weight heatmaps referenced in Section 4.2 of the main paper. These include before/after training visualizations for FMoW (2016 test year) and DomainNet ("sketch" target domain) after training the hypernetwork for 5 epochs.

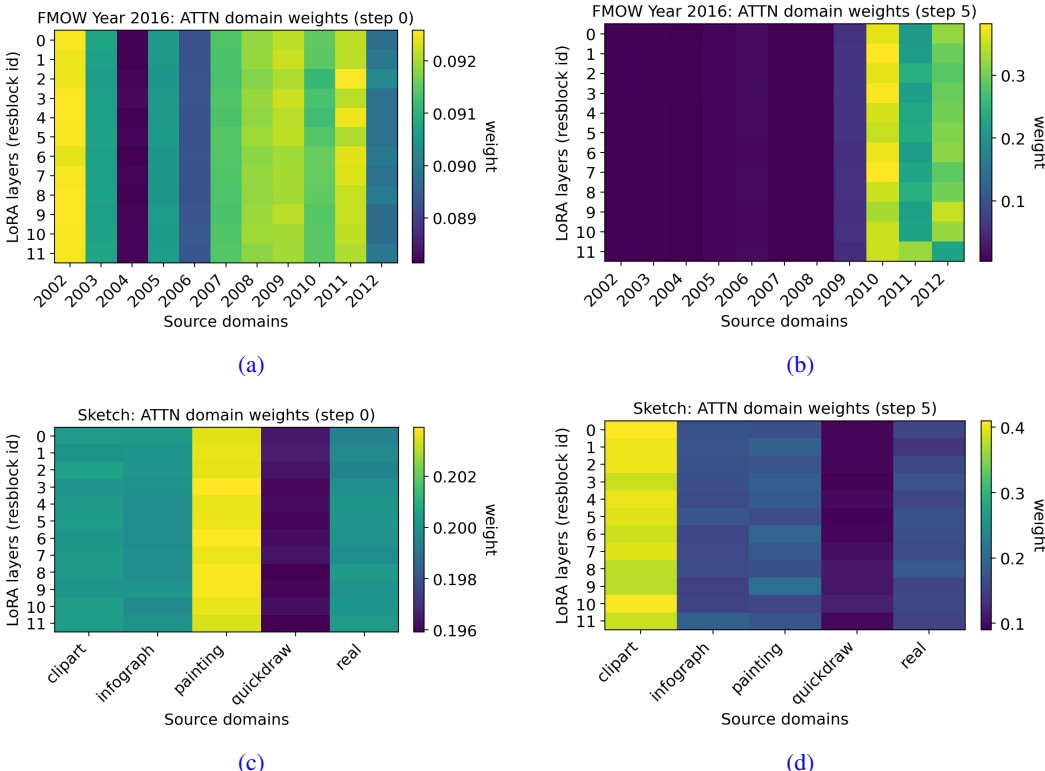

Figure 2: Heatmaps of average merge weights - These figures illustrate the average merge weights per-layer produced by the hypernetwork for each of the source domains. The x-axis shows the source domains and the y-axis shows the transformer layer. Plots (a) and (c) show the merge weights before training on FMoW (domain - Year 2016) and DomainNet (domain - Sketch). Plots (b) and (d) show results after training. Before training, merge weights are nearly uniform across source domains. After training, the hypernetwork shifts merge weights towards source domains visually similar to the target domain, and down-weights dissimilar and difficult domains.

### A.3.2 MLP vs. Cross-Attention Hypernetwork Architectures

We compare our hypernetwork architecture against an MLP-based hypernetwork. Both architectures have a similar number of parameters (MLP: ∼690k; cross-attention: ∼600k). The MLP concatenates the target image embedding, layer tokens, domain tokens, and LoRA columns as input, and consists of two layers with a hidden size of 256. Results are shown in Appendix Table 4. Results show that both architectures achieve comparable performance. In general, MLPs may provide more stable training, but cross-attention is better suited for modeling inter-domain relationships. We expect future work exploring richer cross-attention variants (multi-head or bidirectional) to be more promising than simply deepening MLPs, as cross-attention is better at capturing complex dependencies.

Table 4: Ablation of multi-layer perceptron (MLP) vs. cross-attention (CA) hypernetwork architectures. Results are reported as mean (standard deviation) over three trials with different random seeds. The † symbol denotes the official dataset metric.

| Method | IWildCam Acc | IWildCam Macro F1† | Camelyon17 Acc† | FMoW WC Acc† | FMoW Acc |
|--------|-------------|--------------------|-----------------|--------------|----------|
| MLP | 71.97 (2.41) | 31.83 (2.12) | 96.85 (0.12) | **53.17 (0.10)** | **57.99 (0.42)** |
| CA (ours) | **73.70 (1.49)** | **33.62 (1.63)** | **96.90 (0.14)** | 52.26 (0.16) | 57.60 (0.14) |

### A.3.3 EFFECTS OF DIFFERENT SUPPORT SET BATCH SIZES

We assess our hypernetwork's robustness to different batch sizes for the unlabeled target support set, which is used to guide the merging process. We evaluate batch sizes of 1, 4, 8, and 16 (ours), and also include a baseline with no conditional input for reference, where a matrix of zeros is provided to the hypernetwork. During source-domain meta-training, we use a batch size of 16 for both the query and support sets. Results are shown in Appendix Table 5. Performance remains consistent across metrics (within $< 1\%$ variation), except for a slight drop on IWildCam when using a batch size of 1 ($\sim 3\%$ decrease), demonstrating that our method performs meaningful merging even with very limited target data.

Table 5: Comparison of different support set batch sizes at inference time for our algorithm DA-MergeLoRA. Results are reported as mean (standard deviation) over three trials with different random seeds. The † symbol denotes the official dataset metric.

| Method | IWildCam Acc | IWildCam Macro F1† | Camelyon17 Acc† | FMoW WC Acc† | FMoW Acc |
|--------|-------------|--------------------|-----------------|--------------|----------|
| Batch size 0 (zeros as input) | 55.72 (0.99) | 17.35 (0.51) | 95.87 (0.49) | 43.06 (0.27) | 48.40 (0.12) |
| Batch size 1 | 71.32 (0.55) | 29.99 (1.34) | 96.91 (0.13) | **52.30 (0.25)** | **57.62 (0.21)** |
| Batch size 4 | 73.64 (1.36) | 32.91 (2.05) | **96.91 (0.14)** | 52.24 (0.16) | 57.60 (0.13) |
| Batch size 8 | **73.71 (1.40)** | 33.47 (1.96) | 96.90 (0.14) | 52.26 (0.17) | 57.60 (0.14) |
| Batch size 16 (ours) | 73.70 (1.49) | **33.62 (1.63)** | 96.90 (0.14) | 52.26 (0.16) | 57.60 (0.14) |

### A.3.4 EMBEDDING-SPACE VISUALIZATION (T-SNE)

To assess whether the merged LoRA produces meaningful domain-specific features, we compare t-SNE visualizations of target-domain image embeddings generated by the baseline CLIP model versus the merged domain-specific LoRA model. We compute image embeddings for 2,000 randomly sampled "painting" DomainNet images spanning various classes. Appendix Fig. 3 shows the plots, where each colour represents a different class. We circle two classes before and after applying the merged LoRA model to illustrate the improved separation. The t-SNE plot for the merged LoRA model exhibits tighter class clusters and inter-class separation, indicating that the merged domain-specific LoRA yields more discriminative feature representations.

### A.3.5 QUALITATIVE PREDICTION EXAMPLES

To better understand why the hypernetwork improves prediction quality, we qualitatively analyze cases where the baseline CLIP model fails but the merged LoRA model succeeds. We include 8 examples from the "painting" domain (Appendix Fig. 4a) and 8 examples from the "quickdraw" domain (Appendix Fig. 4b) of DomainNet. In the painting domain, the baseline model frequently misclassifies images as "picture frame" because it focuses on the border of the artwork rather than the object depicted inside, whereas the merged LoRA model correctly identifies the underlying object. In the quickdraw domain, the baseline CLIP model often misclassifies sketches as "squiggle," failing to recognize the intended object, while the merged LoRA model correctly interprets the concept despite the abstract style.

After observing these qualitative trends, we also evaluated whether these distractor-class errors occur systematically across the full target domains. In quickdraw, the baseline predicts the distractor class squiggle for 19.25% of images, compared to 11.71% for the merged model. In the painting domain, the baseline misclassifies 1.57% of images as picture frame, whereas the merged model reduces this

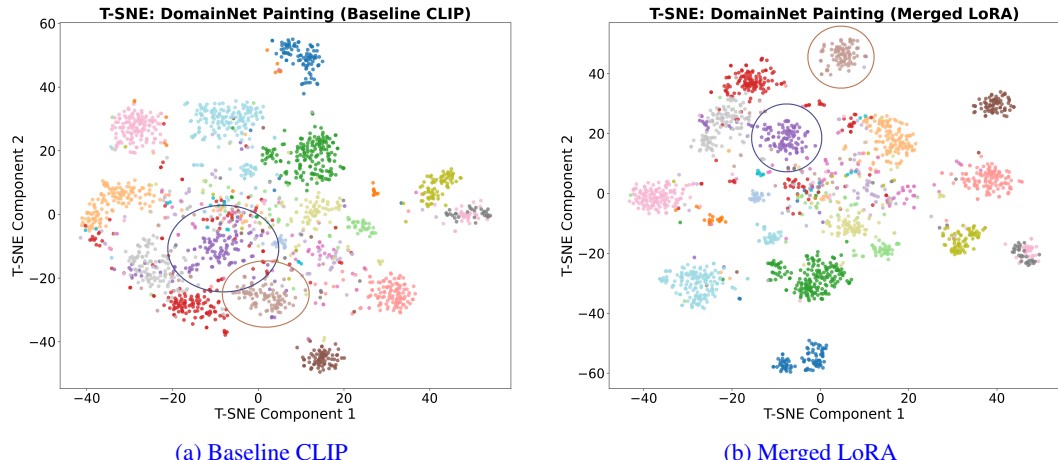

(a) Baseline CLIP         (b) Merged LoRA

Figure 3: T-SNE comparison on the "painting" DomainNet domain for (a) the baseline CLIP model and (b) the merged painting-specific LoRA model, computed over 2,000 images. Each colour represents a different class. We circle two classes before and after applying the merged LoRA model to illustrate the improved separation.

to $0.40\%$. Note that a $1.57\%$ misclassification rate is significant, given that the vast majority of painting-domain images do not contain picture frames.

These examples and statistics demonstrate that the merged LoRA parameters help the model attend to domain-appropriate visual cues and improve reliability under appearance shifts.

### A.3.6 MERGE-WEIGHT VALUES VS. DOMAIN SIMILARITY - CORRELATION

To analyze the relationship between source domains and the predicted merge weights, we plot the average merge weight assigned to each source domain against the cosine similarity between that source domain and the target domain after training the hypernetwork for 5 epochs. We compute cosine similarity using 16 image embeddings per domain, extracted with the baseline CLIP model. We present results for the IWildCam target domain "29" and for the DomainNet "Real" domain. As shown in Appendix Fig. 5, there is a positive correlation: visually similar domains (those with higher cosine similarity to the target) receive higher merge weights. This indicates that the learned merging policy aligns with intuitive domain relationships. Another factor that influences the merge weights is source-domain difficulty: domains that achieve lower accuracy during meta-training tend to be down-weighted. For example, "quickdraw," (the most challenging DomainNet domain) receives the lowest merge weight.

### A.3.7 COMPUTATIONAL ANALYSIS

Below we analyze the memory footprint and computational requirements of our method.

**Memory.** The ViT-B/16 CLIP vision encoder has $\sim 87$M parameters (out of $\sim 150$M total in CLIP, though only the vision encoder is used). Each LoRA module adds 884,736 parameters ($< 1\%$ of the base model), and the hypernetwork adds $\sim 600$k parameters. After merging, only the base model with the synthesized target-domain LoRA is required; source LoRAs and the hypernetwork are not used.

**Test-time computation.** The hypernetwork is executed once to generate merge weights and construct the target-domain LoRA. This one-time operation takes $\sim 1.76$ seconds on average across datasets. Afterward, inference is identical to a standard LoRA-augmented CLIP model, with no additional runtime overhead.

**Comparison with other methods.** Appendix Table 6 compares our train and inference time peak memory usage (MB) and speed (seconds per batch) against the results reported in the L2C-CPNet framework (Chi et al., 2025). Our method requires roughly $50\%$ less memory and is $50\%$ faster at

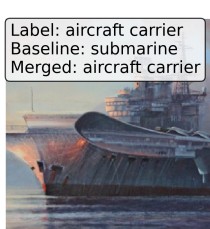
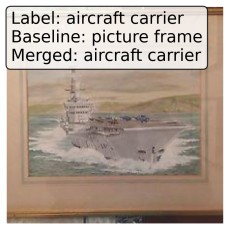
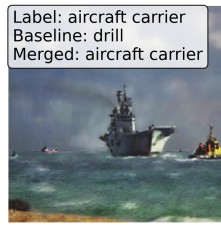
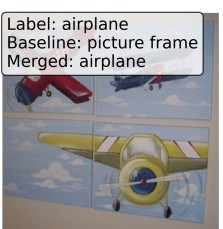

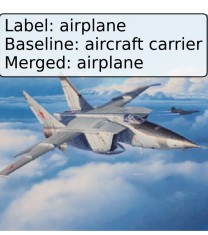
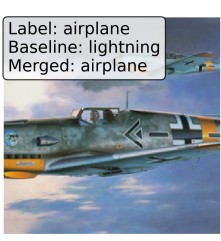
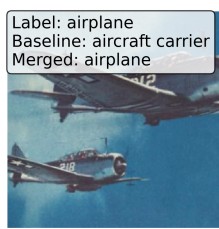
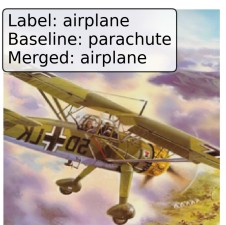

(a) Examples from the "painting" target domain of DomainNet where the baseline CLIP model misclassifies the image, while the merged model predicts the correct label.

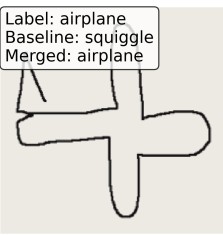
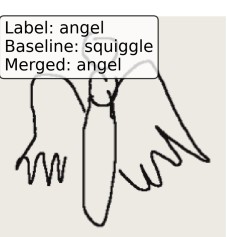
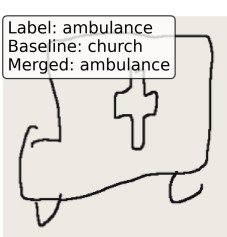
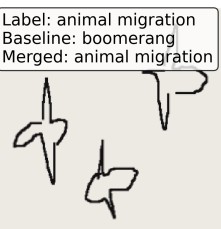

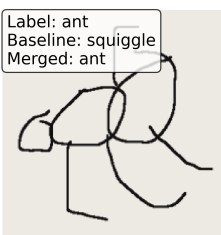
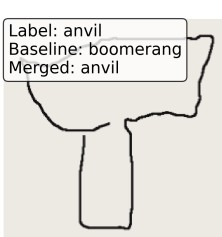
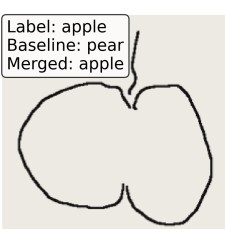
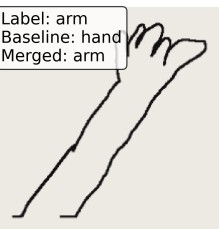

(b) Examples from the "quickdraw" target domain of DomainNet where the baseline CLIP model misclassifies the image, while the merged model predicts the correct label.

Figure 4: Qualitative comparison illustrating how the merged LoRA model improves predictions over the baseline CLIP model across various DomainNet domains.

inference time compared to previous approaches, making it advantageous for FSTT-DA deployment. Our training memory is comparable to prior methods, with slightly slower training speed. Since meta-training occurs offline before deployment, we assume sufficient pre-deployment compute as in standard TTA and meta-learning pipelines.

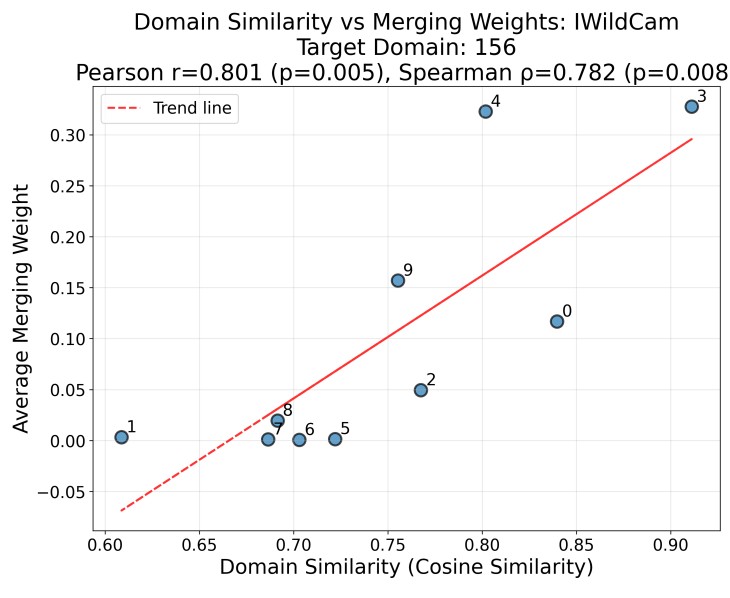

(a) IWildCam Target domain: 29

(b) DomainNet Target domain: Real

Figure 5: Correlation between domain similarity and average merging weights - To analyze the relationship between source domains and the predicted merge weights, we plot the average merge weight assigned to each source domain against the cosine similarity between that source domain and the target domain after training.

Table 6: Training and Inference-time speed (batch per second) and memory usage (MB) compared to the L2C-CPNet (Chi et al., 2025) and VDPG (Chi et al., 2024) FSTT-DA frameworks. N denotes the number of source domains.

| Method | CPNet (1 layer) | CPNet (3 layer) | CPNet (6 layer) | VDPG | DA-MergeLoRA |
|---|---|---|---|---|---|
| Datasets | Camelyon17 | DomainNet, iWildCam | FMoW | All datasets | All datasets |
| # Learnable params | 13.8M | 27.9M | 49.2M | 32.1M | $600k + 885k \times N$ |
| Train memory (MB) | 3872 | 5554 | 8106 | 3672 | 4739 |
| Train speed (s/batch) | 0.84 | 0.88 | 0.97 | 0.87 | 1.78 |
| Inference memory (MB) | 2270 | 2330 | 2430 | 2798 | 1056 |
| Inference speed (s/batch) | 0.64 | 0.67 | 0.73 | 0.69 | 0.34 |

