# OpenReview forum: "DA-MergeLoRA: Hypernetwork-Based LoRA Merging for Few-Shot Test-Time Domain Adaptation"
_ICLR.cc/2026/Conference — Submitted to ICLR 2026_

### Official Review · Reviewer_gQbd · 2025-10-31

**Soundness:** 3
**Presentation:** 2
**Contribution:** 2
**Rating:** 4
**Confidence:** 3

**Summary:**

This paper tackles the problem of few-shot test-time domain adaptation.
The authors train a LoRA adapter for each source domain on top of a fixed CLIP backbone model.
They then introduce a hypernetwork that determines how to combine these adapters with appropriate weights.
The developed model demonstrates superior performance compared to baseline methods across various datasets.

**Strengths:**

The authors propose a straightforward and simple strategy which can be an effective solution.
I also appreciate the approach of using LoRA to learn the specific characteristics of each source domain while effectively preserving the backbone knowledge, and then combining them using hypernetwork.
The performance improvement over existing methods appears to be clear.
I think ablation study is thorough convincingly demonstrating the advantages of each component.

**Weaknesses:**

The main weakness of this paper lies in its presentation.

In my opinion, the paper is written in a way that shows little consideration for the reader in many parts.

- The font size in the figures is too small to be readable after printing.
- The explanations for the main tables are insufficient, forcing readers to repeatedly refer back to the details to fully understand them. In lines 368–369, when reporting results, the authors describe the results of the datasets in reverse order—from DomainNet to iWildCam—which makes it difficult to follow immediately.
- Even when reading the Introduction and Related Work sections, the key ideas that differentiate this paper are not clearly or immediately conveyed.
- There are no qualitative examples at all, and only quantitative evaluations are presented, making it hard to grasp how the proposed method actually works in practice.
- Furthermore, beyond benchmarking, the paper lacks discussion or demonstrations of how the proposed method could be **applied** in practice. For example, it would be convincing to show qualitative examples illustrating that the model generalizes well to the sketch domain even without training on sketch data—but such demonstrations are missing.

Overall, while I think the paper’s **straightforward and simple approach** is a strength, the **practical advantages** are not clearly shown, making the **contribution feel weak**. For these reasons, I am leaning toward the **reject** side.

**Questions:**

At test time, does the model need the 16 target-domain images (batch size) and a target image?

In some cases, the target domain may be extremely limited, containing only one to three images. I would like to know how robust the model is to variations in the number of available target-domain samples.

---

> ### Author Response · Authors · 2025-11-25
> **Response to Question 1 and 2**
>
> We thank the reviewer for their feedback and suggestions. We address all concerns in the comments below.
>
> ## Q1
> > Q1. The main weakness of this paper lies in its presentation. In my opinion, the paper is written in a way that shows little consideration for the reader in many parts. The font size in the figures is too small to be readable after printing. The explanations for the main tables are insufficient, forcing readers to repeatedly refer back to the details to fully understand them. In lines 368–369, when reporting results, the authors describe the results of the datasets in reverse order—from DomainNet to iWildCam—which makes it difficult to follow immediately.
>
> We have added the following changes to address these concerns:
> - We have increased the font sizes in all figures (Fig. 1 and Fig. 2) for better readability. Due to space limits, the full merging-weight heatmaps are now shown in the appendix at a larger and clearer scale (Appendix A.3.1, lines 810-851).
> - We have expanded the explanations accompanying the main tables and figures (lines 432-465, 486-515, and 819-851) to make them more self-contained.
> - We have reordered the reported dataset results so the progression matches the table layout and is easier to follow (lines 376-392).
>
> ## Q2
> > Q2. When reading the Introduction and Related Work sections, the key ideas that differentiate this paper are not clearly or immediately conveyed.
>
> Note: Please see our responses to Reviewer oEDv for questions Q5 and Q6 for a detailed, paper-by-paper comparison with prior LoRA-merging and hypernetwork-based methods, where we address this concern in depth and indicate the specific sections of the paper that were revised for clarity.
>
> While our method builds on established components, our approach fundamentally differs from previous work. Typical model-merging techniques are designed to build generalizable multi-task models. In contrast, our goal is to build specialized, domain-specific models. Our key novelty lies in conditioning the LoRA-merging process on target-domain features, extracted from a few unlabeled target images at test time. Unlike diffusion-based hypernetwork approaches, our hypernetwork accepts two modalities: the LoRA models (as in prior work) and a batch of target images that guide the merge. This conditioning allows the hypernetwork to generate domain-specialized merge weights, enabling one-shot, domain-specific adaptation.

---

> ### Author Response · Authors · 2025-11-25
> **Response to Question 3**
>
> ## Q3
> > Q3. There are no qualitative examples at all, and only quantitative evaluations are presented, making it hard to grasp how the proposed method actually works in practice. Furthermore, beyond benchmarking, the paper lacks discussion or demonstrations of how the proposed method could be applied in practice. For example, it would be convincing to show qualitative examples illustrating that the model generalizes well to the sketch domain even without training on sketch data—but such demonstrations are missing.
>
> In the revised version, we include three forms of qualitative analysis to illustrate how DA-MergeLoRA behaves in practice and why it improves adaptation. Due to page limits, these additions are provided in the Appendix (A3.4, A3.5, and A3.6, lines 894-1076).
>
> **Qualitative prediction examples (lines 904-917, 939-943):**
> - To better understand why the hypernetwork improves prediction quality, we qualitatively analyze cases where the baseline CLIP model fails but the merged LoRA model succeeds. We include 8 examples from the "painting" domain and 8 from the "quickdraw" domain of DomainNet (lines 972-1022). In the painting domain, the baseline model frequently misclassifies images as "picture frame" because it focuses on the border of the artwork rather than the object depicted inside, whereas the merged LoRA model correctly identifies the underlying object. In the quickdraw domain, the baseline CLIP model often misclassifies sketches as "squiggle," failing to recognize the intended object, while the merged LoRA model correctly interprets the concept despite the abstract style. These examples demonstrate that the merged LoRA parameters help the model attend to domain-appropriate visual cues and improve reliability under appearance shifts.
>
> **Merge-weight values vs. domain similarity correlation (lines 944-956):**
> - To analyze the relationship between source domains and the predicted merge weights, we plot the average merge weight assigned to each source domain against the cosine similarity between that source domain and the target domain. We compute cosine similarity using 16 image embeddings per domain, extracted with the baseline CLIP model. We present results for the IWildCam target domain “29” and for the DomainNet “Real” domain. As shown in the figure (lines 1029-1076), there is a positive correlation: visually similar domains (those with higher cosine similarity to the target) receive higher merge weights. This indicates that the learned merging policy aligns with intuitive domain relationships. Another factor that influences the merge weights is source-domain difficulty: domains that achieve lower accuracy during meta-training tend to be down-weighted. For example, “quickdraw,” (the most challenging DomainNet domain) receives the lowest merge weight.
>
> **Embedding-space visualization (t-SNE) (lines 893-903):**
> - To assess whether the merged LoRA produces meaningful domain-specific features, we compare t-SNE visualizations of target-domain embeddings generated by the merged model versus the baseline CLIP model. We compute embeddings for 2,000 randomly sampled “painting” domain images from DomainNet. The t-SNE plot (lines 918-937) for the merged model shows significantly tighter class clusters and cleaner separation, indicating that the merged domain-specific LoRA yields more discriminative feature representations.

---

> ### Author Response · Authors · 2025-11-25
> **Response to Questions 4 and 5**
>
> ## Q4
> > Q4. Overall, while I think the paper’s straightforward and simple approach is a strength, the practical advantages are not clearly shown, making the contribution feel weak. For these reasons, I am leaning toward the reject side.
>
> We have strengthened the paper to more clearly demonstrate the practical advantages of DA-MergeLoRA. As detailed in our response to Q3, we now include qualitative analyses, embedding-space visualizations, and merge-weight interpretability studies that together show how the method adapts a pretrained model to an unseen domain using only a few unlabeled target samples. These additions clarify how the method behaves in practice and provide intuitive evidence of its effect beyond quantitative metrics.
>
> ## Q5
> > Q5. At test time, does the model need the 16 target-domain images (batch size) and a target image? In some cases, the target domain may be extremely limited, containing only one to three images. I would like to know how robust the model is to variations in the number of available target-domain samples.
>
> Yes, our method uses a batch size of 16 at test time for the target support set. We have added an additional ablation study in Appendix A3.3.3 that assesses the impact of different batch sizes (lines 872–892), summarized below.
>
> **Batch Size Analysis:**
> - We assess our hypernetwork’s robustness to different batch sizes for the unlabeled target support set, which is used to guide the merging process. We evaluate batch sizes of 1, 4, 8, and 16 (ours), and also include a baseline with no conditional input for reference, where a matrix of zeros is provided to the hypernetwork. During source training, we use a batch size of 16 for both the query and support sets. Our method remains robust across different batch sizes, with results staying consistent (less than 1% difference) except for a slight drop on IWildCam when using a batch size of 1 (3.6% reduced F1-score) , showing that our method is able to perform meaningful merging even under limited target data.
>
> | **Method**                 | **IWildCam Acc** | **IWildCam Macro F1†** | **Camelyon17 Acc†** | **FMoW WC Acc†** | **FMoW Acc** |
> |---------------------------|------------------|--------------------------|----------------------|-------------------|---------------|
> | Batch size 0 (zeros as input) | 55.72 (0.99)     | 17.35 (0.51)             | 95.87 (0.49)         | 43.06 (0.27)      | 48.40 (0.12)  |
> | Batch size 1              | 71.32 (0.55)     | 29.99 (1.34)             | 96.91 (0.13)         | **52.30 (0.25)**  | **57.62 (0.21)** |
> | Batch size 4              | 73.64 (1.36)     | 32.91 (2.05)             | **96.91 (0.14)**     | 52.24 (0.16)      | 57.60 (0.13)  |
> | Batch size 8              | **73.71 (1.40)** | 33.47 (1.96)             | 96.90 (0.14)         | 52.26 (0.17)      | 57.60 (0.14)  |
> | Batch size 16 (ours)      | 73.70 (1.49)     | **33.62 (1.63)**         | 96.90 (0.14)         | 52.26 (0.16)      | 57.60 (0.14)  |

---

### Official Review · Reviewer_D3hq · 2025-10-31

**Soundness:** 3
**Presentation:** 3
**Contribution:** 3
**Rating:** 6
**Confidence:** 4

**Summary:**

This paper addresses the problem of Few-Shot Test-Time Domain Adaptation problem, where models must adapt to unseen target domains using only a few unlabeled samples. The authors propose a framework that combines LoRA modules trained on source domains into a single target-specific representation via a hypernetwork-based merging strategy. This hypernetwork, trained using meta-learning, predicts per-column merging weights to fuse LoRA modules conditioned on target-domain samples. The approach aims to address the limitations of existing FSTT-DA methods, such as shallow updates, restricted knowledge transfer, and inefficiencies. The paper claims state-of-the-art performance on several benchmarks.

**Strengths:**

1. The Few-Shot Test-Time Domain Adaptation setting reflects real-world challenges, where target domain data is scarce and unlabeled. The proposed approach aligns well with this problem and offers a potentially efficient solution through LoRA merging.

2. By leveraging LoRA, which modifies only a small subset of model parameters, the method achieves parameter-efficient adaptation. The use of a hypernetwork for merging LoRA modules provides a novel mechanism for domain adaptation without requiring full fine-tuning or ensembling.

3. The paper evaluates DA-MergeLoRA across multiple datasets and compares it to a range of baselines, including SOTA FSTT-DA methods.

4. The parameter-space merging formulation is grounded in prior work on LoRA and model merging, and the use of per-column merging with cross-attention is a reasonable design for fine-grained adaptation.

5. The paper is well written and easy to read.

**Weaknesses:**

1. While the combination of LoRA and hypernetwork-based merging is novel in the context of FSTT-DA, the individual components (LoRA fine-tuning, hypernetwork-based weight generation, meta-learning) are well-established in prior works. The paper does not demonstrate significant conceptual advances beyond adapting these techniques for FSTT-DA.

2. The method relies heavily on hyperparameters. As shown in sensitivity analyses, improper tuning can degrade performance significantly, limiting the robustness of the approach.

3. The target domains are simulated by masking pseudo-target LoRA modules during training. This setup may not reflect the complexity of real-world domain shifts, where target domains may exhibit more nuanced differences from source domains.

4. The merging weights generated by the hypernetwork are not interpretable. There is no analysis of why certain LoRA modules are weighted more heavily or how the merging process aligns with the target domain characteristics.

**Questions:**

Please refer to the weaknesses above.

---

> ### Author Response · Authors · 2025-11-25
> **Response to Questions 1 and 2**
>
> We thank the reviewer for their feedback and provide responses to all concerns in the comments below.
>
> ## Q1
> > Q1. While the combination of LoRA and hypernetwork-based merging is novel in the context of FSTT-DA, the individual components (LoRA fine-tuning, hypernetwork-based weight generation, meta-learning) are well-established in prior works. The paper does not demonstrate significant conceptual advances beyond adapting these techniques for FSTT-DA.
>
> Note: Please see our responses to Reviewer oEDv for questions Q5 and Q6 for a detailed, paper-by-paper comparison with prior LoRA-merging and hypernetwork-based methods, where we address these concerns in depth and indicate the specific sections of the paper that were revised for clarity.
>
> While our method builds on established components, our approach fundamentally differs from previous work. Typical model-merging techniques are designed to build generalizable multi-task models. In contrast, our goal is to build specialized, domain-specific models. Our key novelty lies in conditioning the LoRA-merging process on target-domain features, extracted from a few unlabeled target images at test time. Unlike diffusion-based hypernetwork approaches, our hypernetwork accepts two modalities: the LoRA models (as in prior work) and a batch of target images that guide the merge. This conditioning allows the hypernetwork to generate domain-specialized merge weights, enabling one-shot, domain-specific adaptation.
>
> ## Q2
> > Q2. The method relies heavily on hyperparameters. As shown in sensitivity analyses, improper tuning can degrade performance significantly, limiting the robustness of the approach.
>
> While our method includes several hyperparameters (LoRA rank, domain-token scaling factor, number of meta-episodes), our results indicate that the approach is not overly sensitive, and we provide the following clarifications.
>
> The main purpose of the component-analysis ablation (Table 2c) is to justify the hypernetwork's architectural design. As shown in the table, removing learnable layer tokens, domain tokens, or the conditional target-image input degrades performance. This demonstrates that these components provide useful information and are essential for conditional model merging, which is one of our core contributions. For example, when conditional images are removed, the hypernetwork no longer receives domain-specific signals, and the expected drop in performance confirms that our model is indeed performing target-conditioned merging rather than learning a generic merge. Likewise, removing layer or domain tokens weakens the hypernetwork’s ability to reason over site- and domain-specific LoRA structure, validating their necessity.
>
> Additionally, many hyperparameters are simply architectural choices (LoRA rank, hidden sizes, normalization, batch sizes) that are standard across LoRA-based and meta-learning approaches. We keep the same hypernetwork architecture and hyperparameters across all datasets, except for a single scalar (the domain-token scaling factor for iWildCam). We also use only two learning rates across all experiments (5e-4 for LoRA and 1e-5 or 5e-5 for the hypernetwork). This consistency demonstrates that our method performs robustly without relying on dataset-specific hyperparameter adjustments.
>
> Finally, prior FSTT-DA approaches such as VDPG, Meta-DMoE, and L2C also include several hyperparameters (prompt length, prompt LR, number of meta-iterations, etc.). Likewise, LoRA-merging methods such as ZipLoRA and LoRA.rar require tuning of merging-related coefficients or hypernetwork parameters. Our method does not rely on hyperparameters more than existing approaches in either FSTT-DA or LoRA-merging literature.

---

> ### Author Response · Authors · 2025-11-25
> **Response to Questions 3 and 4**
>
> ## Q3
> > Q3. The target domains are simulated by masking pseudo-target LoRA modules during training. This setup may not reflect the complexity of real-world domain shifts, where target domains may exhibit more nuanced differences from source domains.
>
> Our pseudo-target domain method follows a standard practice in meta-learning and leave-one-domain-out adaptation, where the goal is to simulate domain shifts in a controlled and reproducible way. While this procedure does not capture the full complexity of real-world domain shifts, it does expose the hypernetwork to a diverse set of source-domain combinations and forces the model to generalize across heterogeneous LoRA updates.
>
> We agree that this is a potential limitation and view it as a promising direction for future work. For example, more complex pseudo-target domains could be created through data augmentation, style mixing, or combining images from multiple source domains to increase diversity during meta-training. Exploring these directions would likely further strengthen our approach without modifying the core method.
>
> We have updated our conclusion (lines 537–538) to include this as a potential direction for future research.
>
>
> ## Q4
> > Q4. The merging weights generated by the hypernetwork are not interpretable. There is no analysis of why certain LoRA modules are weighted more heavily or how the merging process aligns with the target domain characteristics.
>
> Our submission currently includes an analysis of the merging weights in Sec. 4.2 (Fig. 2), where we visualize average per-domain weights before and after meta-training. These heatmaps show that the hypernetwork assigns higher weights to source LoRA modules corresponding to domains that are visually similar to the target (example: "clipart" for the "sketch" target domain) and down-weights dissimilar domains, indicating that the merging behavior aligns with meaningful target-domain characteristics rather than arbitrary coefficients.
>
> Additionally, we have added three ablation analyses in the Appendix (A3.4, A3.5, and A3.6, lines 894-1076) to illustrate how the merged LoRA model aligns with the target domain and provide intuition for why certain source LoRA modules receive higher weights:
>
> **Merge-weight values vs. domain similarity correlation (lines 944-956):**
> - To analyze the relationship between source domains and the predicted merge weights, we plot the average merge weight assigned to each source domain against the cosine similarity between that source domain and the target domain. We compute cosine similarity using 16 image embeddings per domain, extracted with the baseline CLIP model. We present results for the IWildCam target domain “29” and for the DomainNet “Real” domain. As shown in the figure (lines 1029-1076), there is a positive correlation: visually similar domains (those with higher cosine similarity to the target) receive higher merge weights. This indicates that the learned merging policy aligns with intuitive domain relationships. Another factor that influences the merge weights is source-domain difficulty: domains that achieve lower accuracy during meta-training tend to be down-weighted. For example, “quickdraw,” (the most challenging DomainNet domain) receives the lowest merge weight.
>
> **Qualitative prediction examples (lines 904-917, 939-943):**
> - To better understand why the hypernetwork improves prediction quality, we qualitatively analyze cases where the baseline CLIP model fails but the merged LoRA model succeeds. We include 8 examples from the "painting" domain and 8 from the "quickdraw" domain of DomainNet (lines 972-1022). In the painting domain, the baseline model frequently misclassifies images as "picture frame" because it focuses on the border of the artwork rather than the object depicted inside, whereas the merged LoRA model correctly identifies the underlying object. In the quickdraw domain, the baseline CLIP model often misclassifies sketches as "squiggle," failing to recognize the intended object, while the merged LoRA model correctly interprets the concept despite the abstract style. These examples demonstrate that the merged LoRA parameters help the model attend to domain-appropriate visual cues and improve reliability under appearance shifts.
>
> **Embedding-space visualization (lines 893-903):**
> - To assess whether the merged LoRA produces meaningful domain-specific features, we compare t-SNE visualizations of target-domain embeddings generated by the merged model versus the baseline CLIP model. We compute embeddings for 2,000 randomly sampled “painting” domain images from DomainNet. The t-SNE plot (lines 918-937) for the merged model shows significantly tighter class clusters and cleaner separation, indicating that the merged domain-specific LoRA yields more discriminative feature representations.

---

### Official Review · Reviewer_oEDv · 2025-10-31

**Soundness:** 2
**Presentation:** 2
**Contribution:** 2
**Rating:** 4
**Confidence:** 5

**Summary:**

The authors proposes DA-MergeLoRA for few-shot test-time domain adaptation (FSTT-DA) using a frozen CLIP ViT‑B/16 vision encoder augmented with LoRA adapters trained per source domain. It combines LoRA fine-tuning and model merging to efficiently adapt pre-trained VLMs like CLIP to target domains using only a few unlabeled target samples. Each source domain is first fine-tuned with its own LoRA module, which captures domain-specific knowledge while keeping the CLIP backbone frozen to preserve generalization. To adapt to a target domain, the method employs a meta-learned hypernetwork that predicts per-column merging weights to fuse the source LoRA modules into a single target-specific adapter. Merging in the parameter space allows efficient cross-domain knowledge transfer without retraining or ensembling. The proposed FSTT-DA method is validated on DomainNet, Camelyon17, FMoW, and iWildCam classification benchmarks, and results show that it  can outperform against SOTA methods.

**Strengths:**

+ Overall, the paper is clearly written, well organized, and easy to follow. It presents a clear motivation and methodology. DA-MergeLoRA trains a LoRA adapter per source domain on a frozen CLIP encoder and, at test time, uses a meta-learned hyper-network conditioned on a small batch of target images to produce per-column merge weights that combine these adapters into a single target-specific model. Parameter-space merging avoiding multi-expert ensembling,
+ The paper formulates FSTT-DA as a parameter-space merging problem, and relies on the hypernetwork to combine LoRA adapters trained on multiple source domains.
+ DA-MergeLoRA represents a cost-effective alternative to traditional fine-tuning or ensembling methods, and that may be generalized to broader adaptation settings.
+ Empirical results show that the proposed DA-MergeLoRA method can outperform SOTA  on diverse datasets (DomainNet, Camelyon17, FMoW, and iWildCam). Ablations validate important models choices, e.g., that conditional merging on target image data can significantly improves adaptation performance.
+ Their code is made available, and will allow the reader to reproduce the results.

**Weaknesses:**

- The practical motivation for FSTT-DA appears limited. Standard TTA already addresses the challenge of adapting models to unseen domains using unlabeled target data, and most TTA methods operate with small batches or even single-sample updates. Framing a separate “few-shot” variant therefore seems largely quantitative rather than conceptually distinct, as it merely constrains the amount of target data without introducing a new adaptation principle. Moreover, the episodic few-shot setup used in current FSTT-DA studies is somewhat artificial, since in realistic deployments models typically encounter continuous or sufficiently large target data streams. As a result, the additional architectural complexity introduced by meta-learning or hypernetwork-based approaches in this setting may not be justified by a correspondingly strong real-world need or performance gain.
- Experiments: Claims of SOTA performance are mostly supported by the tables but iWildCam protocol deviates from the main meta-training recipe with this claim (skewed class distributions limits the effectiveness of meta-training.) this argument is weak it should be supported by citation or experimental ablation to prove it.
- The authors highlight some merging techniques in the related work. They should contrast their approach with the latest works.  The latest work discussed is 2023.
- Although the paper positions the proposed method within the few-shot test-time domain adaptation (FSTT-DA) paradigm, its design and operational assumptions align more closely with multi-source unsupervised domain adaptation (MSUDA) or model-fusion approaches. The framework trains separate LoRA adapters for each source domain and learns a hypernetwork that merges these domain-specific adapters into a single model at inference. The “adaptation” at test time combines pretrained source experts through a meta-learned weighting mechanism. There is no gradient-based learning, self-supervised objective, or continual update occurs on the target data. Consequently, the process is closer to multi-source parameter aggregation or mixture-of-experts fusion than to genuine test-time adaptation, which traditionally involves adapting model parameters online or per-batch during deployment. From this standpoint, the method should have been evaluated against strong multi-source domain adaptation baselines rather than exclusively against few-shot or prompt-based TTA methods.
- The proposed DA-MergeLoRA has limited novelty. Actually, I was not able identify any clear technical novelty in this work.The authors follows a similar merging technique and weights prediction as LoRA-RaR (Shenaj 2024, Yadav 2023), and expand on meta-learning-based TTA (Zhong 2022; Chi 2024).  I would encourage the authors to clarify the novelty of their contribution relative to ZipLoRA (ECCV 2024) and LoRA.rar (arXiv 2025). ZipLoRA already introduced per-column merging of LoRA adapters through optimization of column-wise coefficients. LoRA.rar extended this idea by using a hypernetwork to predict those coefficients directly, enabling one-shot merging for unseen pairs. The proposed framework appears to adopt an almost identical formulation, which is  a hypernetwork that outputs per-column merge weights, though applied to CLIP rather than diffusion models. Unless there are substantial algorithmic differences, the claimed “per-column hypernetwork-based merging” may not represent a novel contribution. I would appreciate a clearer discussion explaining what aspects are new beyond these prior works.
- The authors’ own ablation studies reveal no advantage of the proposed per-column merging strategy over simpler per-layer alternative. The paper attributes this marginal gain to column misalignment among independently trained LoRA adapters and to the limited expressiveness of the single-head cross-attention hypernetwork. However, this explanation substantially weakens the paper’s claimed contribution. Furthermore, the paper omits direct comparisons with several closely related LoRA-merging frameworks, including MoLE (paper: Mixture of LoRA Experts), LoRAHUB, ZipLoRA, LoRA.rar, KnOTS(paper: Model merging with SVD to tie the Knots). Both ZipLoRA and LoRA.rar already implement column-wise LoRA merging (the latter using a hypernetwork for one-shot coefficient prediction), while MoLE, KnOTS , and LoRAHub address alignment, gating, and interference across LoRAs in more principled ways. Evaluating against these established baselines would have provided a much stronger and fairer assessment of the proposed approach’s effectiveness. In their absence, the contribution appears unsignificant, with limited evidence that the proposed merging granularity or architecture yields genuine improvements beyond existing LoRA fusion techniques.
- The authors do not discuss the computational or memory implications of multiple multiple domain-specific LoRA experts and a meta-trained hypernetwork. These factors that are central in real-world test-time adaptation, where latency and resource budgets are critical. Test-time adaptation methods are typically valued for their on-the-fly adaptability under strict time and memory constraints, whereas the proposed pipeline presupposes the availability of all source adapters and an additional inference-time hypernetwork, which may not be feasible in deployment. It is important to provide runtime or memory comparisons to justify the practicality of this approach in a genuine test-time setting. Design choice of the hypernetwork architecture (single-head cross-attention) are not analyzed. This paper should also contain an experimental analysis of time and memory complexity for hypernetwork training.
 -  Results in Table2c show a large decline. It’s hard to disentangle conditional merging helps from random, misscaled inputs hurt. I suggest removing the target-embedding branch entirely and retrain the hypernetwork under that setting, so capacity and training dynamics are comparable.

References:
ZipLoRA — Shah et al. ZipLoRA: Any Subject in Any Style by Effectively Merging LoRAs. ECCV 2024.
LoRA.RaR — Shenaj et al. LoRA.rar: Learning to Merge LoRAs via Hypernetworks for
Subject-Style Conditioned Image Generation. ICCV 2025.
Task Arithmetic — Ilharco et al. Editing Models with Task Arithmetic. ICLR 2023.
TIES‑Merging — Yadav et al. Resolving Interference When Merging Models. NeurIPS 2023.
KnOTS — Stoica et al. Model merging with SVD to tie the Knots. ICLR 2025.
Iso‑C / Iso‑CTS — Marczak et al. No Task Left Behind: Isotropic Model Merging with Common and Task-Specific Subspaces. ICML 2025.
TSV‑M — Gargiulo et al. Task Singular Vectors: Reducing Task Interference in Model Merging. CVPR 2025.

**Questions:**

See my comments in weaknesses.  In my opinion, the novelty is limited, combining existing LoRA merging and meta-learning methods, while missing a detailed analysis of the hypernetwork design.

**Details Of Ethics Concerns:**

None.

---

> ### Author Response · Authors · 2025-11-25
> **Response to Question 1**
>
> We thank the reviewer for their constructive feedback and address all points in the comments below.
>
> ## Q1
> > Q1. The practical motivation for FSTT-DA appears limited. Standard TTA already addresses the challenge of adapting models to unseen domains using unlabeled target data, and most TTA methods operate with small batches or even single-sample updates. Framing a separate “few-shot” variant therefore seems largely quantitative rather than conceptually distinct, as it merely constrains the amount of target data without introducing a new adaptation principle. Moreover, the episodic few-shot setup used in current FSTT-DA studies is somewhat artificial, since in realistic deployments models typically encounter continuous or sufficiently large target data streams. As a result, the additional architectural complexity introduced by meta-learning or hypernetwork-based approaches in this setting may not be justified by a correspondingly strong real-world need or performance gain.
>
> Regular TTA typically operates under two settings (online and offline). Online TTA adapts the model continuously as target samples arrive, using per-sample or mini-batch updates that accumulate stable statistics such as BatchNorm estimates or entropy-based updates. Offline TTA adapts the model after deployment by performing multi-epoch unsupervised fine-tuning on the entire unlabeled target set. Both settings can be unrealistic in practice. Online TTA introduces continuous computational overhead and can slow inference due to per-sample gradient updates. Offline TTA requires access to a large amount of target data and enough compute to run an unsupervised training loop at deployment time. Furthermore, continually updating the model using target-domain data may be infeasible due to privacy restrictions or safety concerns.
>
> FSTT-DA performs adaptation using a single batch of unlabeled target images. The adapted model is subsequently used for all target-domain predictions, with no iterative fine-tuning required. Thus, FSTT-DA avoids (i) the continual updates required by online TTA, (ii) the need for large-scale target data and multi-step adaptation required by offline TTA, and (iii) any increase in inference-time cost. This makes it highly practical for deployment when target data are scarce, when devices are resource-constrained, or when adaptation must not slow down inference.
>
> Such few-shot conditions arise naturally in real deployments: when a model is introduced in a new hospital (Camelyon17), a wildlife camera trap is deployed to a new location (iWildCam), a satellite imaging a new geographic region (FMoW), or a robot entering an unfamiliar environment. In these scenarios, collecting large target datasets for offline TTA or performing continuous online updates is often infeasible due to privacy constraints, slow data accumulation, limited connectivity, or resource-constrained hardware. However, a few data samples of the new domains can be readily available during inference (data collected during the camera calibration stage). FSTT-DA therefore addresses a distinct and practically relevant setting where the model must perform a single, stable, low-cost adaptation step using only a small initial batch of images.
>
> We have revised our Introduction (lines 34–56) to clarify our problem setting, differentiate FSTT-DA from standard offline and online TTA, and include practical scenarios where the few-shot test-time setting occurs.

---

> ### Author Response · Authors · 2025-11-25
> **Response to Question 2**
>
> ## Q2
> > Q2. Experiments: Claims of SOTA performance are mostly supported by the tables but iWildCam protocol deviates from the main meta-training recipe with this claim (skewed class distributions limits the effectiveness of meta-training.) this argument is weak it should be supported by citation or experimental ablation to prove it.
>
> We clarify that the primary difficulty on iWildCam is not class imbalance per se, but the fact that iWildCam contains disjoint label sets across source domains. Our meta-training assumes a shared label space (standard in domain adaptation and model-merging frameworks) so that source LoRA modules trained on different domains encode updates for the same semantic outputs. In iWildCam, however, many species appear in only a single source domain. Under this condition, during meta-training, merging LoRA modules becomes closer to a novel-class generalization problem rather than domain adaptation. Prior work has shown that domain adaptation performance degrades when label distributions diverge across domains or when classes appear exclusively in one domain (Chidlovskii, 2019; Yan et al., 2017; Tan et al., 2020). This supports our interpretation that the dataset structure makes iWildCam a challenging setting for merging-based DA methods. We have updated our paper (lines 386–392) to expand our explanation and to provide citations supporting our claims. Note that this issue arises only during meta-training, where we simulate target domains by holding out one source domain as the pseudo-target. This does not occur during actual evaluation, where the real target domain’s classes always appear in at least one of the source domains.
>
> Also note that the only procedural difference in the meta-training stage for iWildCam, compared to the other datasets, is that we do not mask the pseudo-target domain. Both masking and non-masking are valid choices for our algorithm: masking simulates encountering an unseen domain at test-time and helps with generalization, while non-masking provides a stronger learning signal and ensures that all classes remain represented during meta-training. In iWildCam, this non-masking strategy is beneficial because it prevents the meta-learning episodes from degenerating into a novel-class generalization problem caused by the dataset’s class-exclusive domains. This is a design choice of the algorithm and does not invalidate the meta-learning setup.
>
> Finally, we note that other CLIP-based FSTT-DA methods (L2C, VDPG) also underperform relative to MABN, suggesting that this behavior is not specific to our approach. A likely contributing factor is the difference in backbones: CLIP is pretrained on internet images, whereas MABN’s ResNet-50 backbone is ImageNet-pretrained and includes many animal-related classes more similar to iWildCam.
>
> References:
> [1] Boris Chidlovskii. Using latent codes for class imbalance problem in unsupervised domain adaptation. arXiv preprint arXiv:1909.08962, 2019.
> [2] Hongliang Yan, Yukang Ding, Peihua Li, Qilong Wang, Yong Xu, and Wangmeng Zuo. Mind the class weight bias: Weighted maximum mean discrepancy for unsupervised domain adaptation. In Proceedings of the IEEE conference on computer vision and pattern recognition, pp. 2272–2281, 2017.
> [3] Shuhan Tan, Xingchao Peng, and Kate Saenko. Class-imbalanced domain adaptation: An empirical odyssey. In European Conference on Computer Vision, pp. 585–602. Springer, 2020.

---

> ### Author Response · Authors · 2025-11-25
> **Response to Questions 3 and 4**
>
> ## Q3
> > Q3. The authors highlight some merging techniques in the related work. They should contrast their approach with the latest works. The latest work discussed is 2023.
>
> We have incorporated several recent merging approaches and updated the related-work section accordingly. A detailed comparison with these methods is provided in our responses to your questions Q5 and Q6, where we clarify the differences in problem setting, assumptions, and applicability; and indicate the corresponding sections of the paper that have been revised.
>
>
> ## Q4
> > Q4. Although the paper positions the proposed method within the few-shot test-time domain adaptation (FSTT-DA) paradigm, its design and operational assumptions align more closely with multi-source unsupervised domain adaptation (MSUDA) or model-fusion approaches. The framework trains separate LoRA adapters for each source domain and learns a hypernetwork that merges these domain-specific adapters into a single model at inference. The “adaptation” at test time combines pretrained source experts through a meta-learned weighting mechanism. There is no gradient-based learning, self-supervised objective, or continual update occurs on the target data. Consequently, the process is closer to multi-source parameter aggregation or mixture-of-experts fusion than to genuine test-time adaptation, which traditionally involves adapting model parameters online or per-batch during deployment. From this standpoint, the method should have been evaluated against strong multi-source domain adaptation baselines rather than exclusively against few-shot or prompt-based TTA methods.
>
> We clarify that our method fits the FSTT-DA setting rather than the multi-source UDA setting. In typical multi-source UDA, the unlabelled target domain is accessible during training, and existing MSUDA methods typically require iterating over large amounts of target data for domain alignment, adversarial training, pseudo-labeling, or distribution matching. In contrast, our method never accesses target data during training and receives only a small few-shot batch at test time. This places our method within the test-time adaptation setting. Although our adaptation mechanism does not rely on gradient-based updates, several recent FSTT-DA approaches similarly perform non-gradient adaptation at test time, such as prompt-based approaches (Chi et al., 2024, Chi et al., 2025). For these reasons, the appropriate comparison group is FSTT-DA methods, which operate under the same constraints and resource assumptions as our method.
>
> We have updated our introduction to clarify the distinction between FSTT-DA and MSUDA (Lines 45–49).
>
> References:
> [1] Zhixiang Chi, Li Gu, Tao Zhong, Huan Liu, Yuanhao Yu, Konstantinos N. Plataniotis, and Yang Wang. Adapting to distribution shift by visual domain prompt generation. In Proceedings of the International Conference on Learning Representations (ICLR),
> 2024. URL https://proceedings.iclr.cc/paper_files/paper/2024/file/
> 440f269a4a6b9d51c51b4997963761ff-Paper-Conference.pdf. ICLR 2024.
> [2] Zhixiang Chi, Li Gu, Huan Liu, Ziqiang Wang, Yanan Wu, Yang Wang, and Konstantinos N Plataniotis. Learning to adapt frozen clip for few-shot test-time domain adaptation. In Proceedings of the International Conference on Learning Representations (ICLR), 2025. URL https://openreview.net/forum?id=TD3SGJfBC7. ICLR 2025.

---

> ### Author Response · Authors · 2025-11-25
> **Response to Question 5**
>
> ## Q5
> > Q5. The proposed DA-MergeLoRA has limited novelty. Actually, I was not able identify any clear technical novelty in this work.The authors follows a similar merging technique and weights prediction as LoRA-RaR (Shenaj 2024, Yadav 2023), and expand on meta-learning-based TTA (Zhong 2022; Chi 2024). I would encourage the authors to clarify the novelty of their contribution relative to ZipLoRA (ECCV 2024) and LoRA.rar (arXiv 2025). ZipLoRA already introduced per-column merging of LoRA adapters through optimization of column-wise coefficients. LoRA.rar extended this idea by using a hypernetwork to predict those coefficients directly, enabling one-shot merging for unseen pairs. The proposed framework appears to adopt an almost identical formulation, which is a hypernetwork that outputs per-column merge weights, though applied to CLIP rather than diffusion models. Unless there are substantial algorithmic differences, the claimed “per-column hypernetwork-based merging” may not represent a novel contribution. I would appreciate a clearer discussion explaining what aspects are new beyond these prior works.
>
> While our method uses LoRA merging as a building block, its goal, setting, and mechanisms differ substantially from ZipLoRA, LoRA.rar, and other existing LoRA-merging frameworks. These works focus on style-content fusion in diffusion models, where two LoRAs (a target style LoRA and a target subject LoRA) are merged to generate images combining both attributes. Although these methods can be applied to previously unseen pairs of LoRAs, they are still limited to combining existing LoRAs and do not perform adaptation to a genuinely new domain that lacks a pretrained LoRA. The goal of these methods is to teach the model how to combine existing target LoRA pairs so they contain the attributes of both.
>
> In contrast, our method addresses few-shot test-time domain adaptation for classification, where the model must adapt to a novel target domain, which has no existing LoRA of its own. The key novelty of our work is conditional LoRA merging, where a hypernetwork extracts target-domain features from a batch of unlabelled target images, to predict a domain-specific configuration to merge source domain LoRA models, which may all differ from the target domain. Unlike diffusion-based hypernetwork approaches, our hypernetwork is mutli-modal, accepting two modalities: the LoRA models (as in prior work) and a batch of target images that guide the merge. This image conditioning enables one-shot adaptation to domains never encountered during training. Our setting also introduces new algorithmic challenges specific to domain adaptation-based model merging, including: (1) domain-conditional merging, (2) multi-modal hypernetwork architecture, (3) leave-one-domain-out meta-training, (4) domain-representation learning, and (5) merging K domain-specific LoRAs instead of just pairs.
>
> Finally, techniques such as column alignment or per-column optimization from previous works are orthogonal to our contributions and could be integrated into our framework as complementary improvements. They do not replace the core novelty of test-time, domain-conditioned merging for unseen domains, which is unique to our method.
>
> We have updated the introduction (lines 87–93) and related-work section (lines 185–191) to clarify these differences.

---

> ### Author Response · Authors · 2025-11-25
> **Response to Question 6 (Part 1 of 2)**
>
> ## Q6 (Part 1 of 2)
> > Q6. The authors’ own ablation studies reveal no advantage of the proposed per-column merging strategy over simpler per-layer alternative. The paper attributes this marginal gain to column misalignment among independently trained LoRA adapters and to the limited expressiveness of the single-head cross-attention hypernetwork. However, this explanation substantially weakens the paper’s claimed contribution. Furthermore, the paper omits direct comparisons with several closely related LoRA-merging frameworks, including MoLE (paper: Mixture of LoRA Experts), LoRAHUB, ZipLoRA, LoRA.rar, KnOTS(paper: Model merging with SVD to tie the Knots). Both ZipLoRA and LoRA.rar already implement column-wise LoRA merging (the latter using a hypernetwork for one-shot coefficient prediction), while MoLE, KnOTS , and LoRAHub address alignment, gating, and interference across LoRAs in more principled ways. Evaluating against these established baselines would have provided a much stronger and fairer assessment of the proposed approach’s effectiveness. In their absence, the contribution appears unsignificant, with limited evidence that the proposed merging granularity or architecture yields genuine improvements beyond existing LoRA fusion techniques.
>
> > References: ZipLoRA — Shah et al. ZipLoRA: Any Subject in Any Style by Effectively Merging LoRAs. ECCV 2024. LoRA.RaR — Shenaj et al. LoRA.rar: Learning to Merge LoRAs via Hypernetworks for Subject-Style Conditioned Image Generation. ICCV 2025. Task Arithmetic — Ilharco et al. Editing Models with Task Arithmetic. ICLR 2023. TIES‑Merging — Yadav et al. Resolving Interference When Merging Models. NeurIPS 2023. KnOTS — Stoica et al. Model merging with SVD to tie the Knots. ICLR 2025. Iso‑C / Iso‑CTS — Marczak et al. No Task Left Behind: Isotropic Model Merging with Common and Task-Specific Subspaces. ICML 2025. TSV‑M — Gargiulo et al. Task Singular Vectors: Reducing Task Interference in Model Merging. CVPR 2025.
>
> The reviewer is correct that our ablations show limited gains for per-column merging over per-layer baselines. Importantly, per-column merging is not the core contribution of our work. Our contribution lies in learning to extract domain-specific information from a novel (unseen) target domain and using it to conditionally merge LoRA modules to construct a new target-domain LoRA, not in defining a new merging granularity.
>
> Below we clarify how prior LoRA-merging works differ from ours:
>
> **Stable-Diffusion LoRA merging (ZipLoRA, LoRA.rar, MoLE)**
>
> - These methods merge existing LoRA modules (typically a style LoRA and a content LoRA) to produce a fused adapter for image generation. Their merging is unconditional, does not depend on target data, and assumes that both LoRAs for the target attributes already exist. In contrast, our method must adapt to a novel target domain with no target LoRA, using only a few unlabeled target images to predict how to combine K source-domain LoRAs into a new domain-specific adapter. Thus, these diffusion approaches solve a fundamentally different problem and cannot perform adaptation to an unseen domain.
>
> **General model-merging frameworks (Task Arithmetic, TIES, KnOTS, Iso-C, Iso-CTS, TSV-M)**
>
> - These methods aim to merge multiple task-specific models into a single general-purpose model while reducing task interference (via SVD, sign alignment, or isotropy, etc.). Their objective is to preserve performance across all original tasks. In contrast, our goal is to obtain a specialized domain-specific model for a novel target domain. Future work could combine our approach with these merging methods: after our hypernetwork identifies domain-relevant LoRA columns, alignment-based techniques (TIES, KnOTS, etc.) could be applied to improve column alignment and reduce parameter interference during merging.
>
> - We have also added an ablation comparing our method to TIES implemented for LoRA merging, where our approach outperforms it (Table 2a). Note that we additionally compare against a universal LoRA model trained on all source data combined; this is a stronger baseline than prior merging methods, as the universal model does not suffer from the parameter-interference or alignment issues that typically degrade the performance of merge-based methods.

---

> ### Author Response · Authors · 2025-11-25
> **Response to Question 6 (Part 2 of 2)**
>
> ## Q6 (Part 2 of 2)
>
> **Task-specific merging in NLP (LoRAHub, MoLE)**
>
> - NLP frameworks such as LoRAHub and MoLE address task adaptation, not domain adaptation. They merge LoRAs trained on different NLP tasks (translation, QA, NLI, etc.) and then adapt to a specific target task. In contrast, our setting keeps the task fixed (classification with the same label space) while the input distribution shifts across visual domains. Domain adaptation requires the model to solve the same task under different visual styles and environments, which is fundamentally different from task adaptation and requires a different algorithmic approach.
>
> - LoRAHub additionally requires labeled target examples and performs multi-step optimization (CMA-ES + cross-entropy) on supervised target data. Our method uses no target labels and adapts in a single forward pass. Applying LoRAHub in our unsupervised, vision-language setting would require redesigning its objective, architecture, and hyperparameters.
>
> - MoLE (NLP) merges LoRAs trained for different tasks using an input-conditioned gating network, allowing the model to switch between task-specific adapters. Its setting assumes that all tasks and their LoRAs are known during training, and adaptation means selecting or combining the appropriate task adapter, not handling shifts in data distribution.
>
> We have updated the introduction (lines 83–98), related-work section (lines 149-161 and lines 182-200), and contributions point #2 (lines 109-113) to clarify the differences between these works and our approach. We have also updated our Table 2a (line 493-494) and "LoRA training techniques" ablation section (lines 416-424) to include TIES for comparison.

---

> ### Author Response · Authors · 2025-11-25
> **Reponse to Question 7**
>
> ## Q7
> > Q7. The authors do not discuss the computational or memory implications of multiple multiple domain-specific LoRA experts and a meta-trained hypernetwork. These factors that are central in real-world test-time adaptation, where latency and resource budgets are critical. Test-time adaptation methods are typically valued for their on-the-fly adaptability under strict time and memory constraints, whereas the proposed pipeline presupposes the availability of all source adapters and an additional inference-time hypernetwork, which may not be feasible in deployment. It is important to provide runtime or memory comparisons to justify the practicality of this approach in a genuine test-time setting. Design choice of the hypernetwork architecture (single-head cross-attention) are not analyzed. This paper should also contain an experimental analysis of time and memory complexity for hypernetwork training.
>
> See our computational and memory analysis summarized below:
>
> - **Memory considerations:** The ViT-B/16 CLIP vision encoder contains 87,077,376 parameters. The total number of CLIP parameters is 150,505,473; however, we only require the vision encoder. Each LoRA module adds only 884,736 parameters (under 1% of the base model). Our hypernetwork adds 600,256 parameters. Even when storing LoRAs from K source domains, the total parameter footprint remains modest compared to storing K full models. Importantly, after the merge is computed, only the base model, with the newly generated target-domain LoRA, needs to be stored; the source LoRAs and hypernetwork are not required during deployment.
>
> - **Computation at test time:** At test time, the hypernetwork performs one forward pass to produce the merge weights and construct the target-domain LoRA. After this one-time operation, inference proceeds identically to a standard LoRA-augmented CLIP model, incurring no additional computational overhead. The one-time adaptation cost to generate and save the merged LoRA model is approximately 1.76 seconds (average across datasets), which is negligible. Also note that the hypernetwork architecture choice has little impact on inference-time, because the hypernetwork is only executed for a one-time forward pass.
>
> - **Comparison:** We add a table comparing our training and inference time speed (batch/s) and peak memory cost (MB) to the results reported in the L2C paper for their algorithm L2C-CPNet and VDPG. Here, N denotes the number of source domains. Our method requires roughly 50% less memory and is 50% faster at inference time compared to previous approaches, making it advantageous for FSTT-DA deployment. Our training memory is comparable to prior methods, with slightly slower training speed. However, note that we did not apply any training-time memory or speed optimizations. Since meta-training occurs offline before deployment, we assume sufficient pre-deployment compute as in standard TTA and meta-learning pipelines.
>
> | **Method**              | **CPNet (1 layer)** | **CPNet (3 layer)** | **CPNet (6 layer)** | **VDPG** | **DA-MergeLoRA** |
> |-------------------------|----------------------|----------------------|----------------------|----------|-------------------|
> | Datasets                | Camelyon17           | DomainNet, iWildCam  | FMoW                 | All datasets | All datasets |
> | # Learnable params      | 13.8M               | 27.9M               | 49.2M               | 32.1M   | 600k + 885k × N |
> | Train memory (MB)       | 3872                | 5554                | 8106                | 3672    | 4739 |
> | Train speed (s/batch)   | 0.84                | 0.88                | 0.97                | 0.87    | 1.78 |
> | Inference memory (MB)   | 2270                | 2330                | 2430                | 2798    | 1056 |
> | Inference speed (s/batch)| 0.64               | 0.67                | 0.73                | 0.69    | 0.34 |
>
> We have added this computational analysis section to the appendix, Section A.3.7 (lines 957–971, 1018-1022, and 1102-1112).

---

> ### Author Response · Authors · 2025-11-25
> **Response to Questions 8 and 9**
>
> ## Q8
>
> > Q8. Results in Table2c show a large decline. It’s hard to disentangle conditional merging helps from random, misscaled inputs hurt. I suggest removing the target-embedding branch entirely and retrain the hypernetwork under that setting, so capacity and training dynamics are comparable.
>
> We have included an additional ablation where we remove the target images and replace them with a trainable query vector. This learnable query vector represents a general merging policy, as opposed to conditional inputs, which enable a domain-specific merging policy. Results also show that the learnable query vector reduces performance less than using noise as input, indicating that a general merging policy can still learn useful priors from the source domains. However, as seen in IWildCam (a decrease of 13.7% in F1-score), for complex domain shifts, a domain-specific model performs better.
>
> We have updated the "Component analysis" ablation section (lines 482–485, 516-526) and the corresponding Table 2c (lines 509–510).
>
> ## Q9
>
> > Q9. Questions: See my comments in weaknesses. In my opinion, the novelty is limited, combining existing LoRA merging and meta-learning methods, while missing a detailed analysis of the hypernetwork design.
>
> We clarify in the revision that our novelty lies in domain-conditioned, one-shot merging of multiple LoRA adapters using only unlabeled target samples, an ability not supported by prior merging or meta-learning approaches. Unlike previous diffusion-based hypernetwork approaches, our hypernetwork accepts two modalities: the LoRA models (as in prior work) and a batch of target images that guide the merge. We also expanded our explanation of the hypernetwork architecture and its design choices. Further details directly addressing these concerns are provided in our previous responses from Q1–Q8.

---

### Author Response · Authors · 2025-11-25
**Thank you to all the reviewers!**

We thank all reviewers for their valuable feedback on our paper regarding hypernetwork-based LoRA merging for few-shot test-time domain adaptation. Reviewers commented that the paper’s approach is easy to follow, methodologically well motivated, and supported by strong empirical results. They also highlighted the effectiveness of our LoRA-based design and our hypernetwork-guided, target-conditioned merging.

Across reviews, the main concerns included clarifying the problem setting, distinguishing our method from existing model-merging approaches, providing more qualitative and interpretability analyses, and improving presentation.

In response, we have made substantial revisions, highlighted in blue in the revised manuscript, including:
1. Clarified the problem setting and positioning of our work in the Introduction and Related Work sections.
2. Added extensive qualitative and interpretability analyses in the appendix, including new visualizations, qualitative examples, additional ablations, and computational analysis.
3. Improved the presentation of the paper by enlarging figure fonts, expanding captions, and reorganizing content.

These updates improve the clarity, evaluation, and presentation of the paper and address the concerns raised by the reviewers.

---

### Author Response · Authors · 2025-12-02
**Rebuttal Summary for the New Area Chair**

Dear Area Chair,

We have made the following changes to the paper to address the reviewers’ concerns:

1. We have updated our Introduction and Related Work sections to clarify our problem setting and contributions, and to better differentiate our method from existing work.
2. We added a direct comparison to TIES model merging in our ablation section.
3. We added extensive qualitative and interpretability analyses in the appendix, including t-SNE visualizations, qualitative examples, domain-similarity vs. merge-weight correlation plots, batch-size analysis, and computational/memory analysis.
4. We improved the presentation of the paper by enlarging figures, expanding captions, and reorganizing sections for readability.

We note that the reviewers have not posted post-rebuttal follow-ups, likely due to the system rollback occurring shortly after our rebuttal submission. However, our rebuttal addresses raised concerns, including comparisons with existing model-merging approaches, clearer differentiation of our method, and additional ablations requested across reviews.

We appreciate your time and consideration.

---

### Meta-Review · Area_Chair_3Yiq · 2026-01-06

**Summary:**

The paper initially received mostly negative scores: 6, 4, 4. The main concerns include: (1) the practical motivation and explanation of FSTT-DA; (2) limited novelty compared to previous works; (3) the need for more ablation studies and analysis; and (4) writing issues. The authors have provided a detailed rebuttal to respond to the reviewers’ concerns.

The AC has carefully read the reviews and the rebuttal, and finds that the authors have addressed concerns (1), (3), and (4), along with other minor points.

However, the critical concern (2) remains insufficiently resolved, namely the novelty relative to prior work. The AC agrees with Reviewers oEDv and D3hq that the proposed method appears to be a combination of well-established techniques, and the significance of this combination for FSTT-DA is not convincingly demonstrated.

Given these considerations, the AC believes the critical concern (i.e., limited novelty) is not adequately addressed in the rebuttal. The AC therefore regretfully recommends rejection.

**Reviewer Concerns:**

Solved Concerns:

Reviewer oEDv: (1) Practical motivation and the statement of FSTT-DA; (2) additional experimental results; (3) computational cost.

Reviewer D3hq: (1) Additional results; (2) explanation of the merging weights.

Reviewer gQbd: (1) Writing issues (partly solved); (2) additional results.

Remaining Concerns:

Reviewers oEDv and D3hq: Limited novelty.

Reviewer D3hq: Heavy reliance on hyper-parameter.

**Reviewer Scores:**

Reviewers oEDv and D3hq may keep their original scores, as the concern regarding limited novelty is not adequately addressed.

While Reviewer D3hq’s specific concerns have been addressed in the rebuttal, he/she may still maintain the original score due to the novelty concern raised by the other reviewers. In addition, as noted by Reviewer D3hq, the presentation should be significantly improved, including clearer explanations of the tables and a more explicit statement of the differences from previous works.

---

### Decision · Program_Chairs · 2026-01-26

Reject